# Monosaccharides drive *Salmonella* gut colonization in a context-dependent or -independent manner

Christopher Schubert [1] ✉, Bidong D. Nguyen[1], Andreas Sichert [2], Nicolas Näpflin [3], Anna Sintsova[1], Lilith Feer [1], Jana Näf[1], Benjamin B. J. Daniel [1], Yves Steiger [1], Christian von Mering[3], Uwe Sauer [2] & Wolf-Dietrich Hardt [1] ✉

The carbohydrates that fuel gut colonization by *S.* Typhimurium are not fully known. To investigate this, we designed a quality-controlled mutant pool to probe the metabolic capabilities of this enteric pathogen. Using neutral genetic barcodes, we tested 35 metabolic mutants across five different mouse models with varying microbiome complexities, allowing us to differentiate between context-dependent and context-independent nutrient sources. Results showed that *S.* Typhimurium uses D-mannose, D-fructose and likely D-glucose as context-independent carbohydrates across all five mouse models. The utilization of D-galactose, *N*-acetylglucosamine and hexuronates, on the other hand, was context-dependent. Furthermore, we showed that D-fructose is important in strain-to-strain competition between *Salmonella* serovars. Complementary experiments confirmed that D-glucose, D-fructose, and D-galactose are excellent niches for *S.* Typhimurium to exploit during colonization. Quantitative measurements revealed sufficient amounts of carbohydrates, such as D-glucose or D-galactose, in the murine cecum to drive *S.* Typhimurium colonization. Understanding these key substrates and their context-dependent or -independent use by enteric pathogens will inform the future design of probiotics and therapeutics to prevent diarrheal infections such as non-typhoidal salmonellosis.

The gut microbiota provides colonization resistance against invading pathogens through diverse mechanisms, including nutrient competition, known as exploitation, and the release of antimicrobial metabolites and type VI secretion systems, termed interference[1]. Despite these mechanisms, *Salmonella enterica* serovar Typhimurium (*S.* Typhimurium), a Gram-negative enteric pathogen, can overcome these defenses, leading to foodborne diarrhea in humans. *S.* Typhimurium is responsible for a considerable number of diarrheal infections[2]. Through animal infection experiments, it has been observed that gut colonization by *S.* Typhimurium occurs in distinct phases[3]. Following oral exposure, *S.* Typhimurium undergoes initial growth without disturbing the resident microbiota[4]. Subsequently, invasion of gut tissue triggers immune responses that decrease the pathogen's tissue burden and alter the gut-luminal environment. *S.* Typhimurium can then flourish in the inflamed gut lumen by utilizing anaerobic and microaerophilic respiration[5–7]. In recent years, some metabolic strategies have been clarified, particularly regarding the types of electron acceptors and some of the electron donors available for

[1]Institute of Microbiology, Department of Biology, ETH Zurich, Zurich, Switzerland. [2]Institute of Molecular Systems Biology, ETH Zurich, Zurich, Switzerland. [3]Department of Molecular Life Sciences and Swiss Institute of Bioinformatics, University of Zurich, Zurich, Switzerland. ✉e-mail: cschubert@ethz.ch; hardt@micro.biol.ethz.ch

*S.* Typhimurium to be utilized[4–23]. Much less is known about carbohydrate utilization. Earlier work found that *S.* Typhimurium uses L-arabinose and the Amadori product fructose-asparagine to proliferate in the inflamed gut[24]. In addition, it has been demonstrated that the hexitol D-galactitol plays a context-dependent role in inter- and intraspecies competition[25,26]. However, a key limitation of this previous research is the predominant use of a single mouse model, which may introduce bias and limit the observable context-dependency of carbohydrate use in these colonization experiments.

Transposon mutagenesis provides a powerful technology to identify mutants with fitness defects during animal infections[27–31]. Genome-wide random-barcoded transposon sequencing (RB-TnSeq) has uncovered metabolic strategies used by *S.* Typhimurium to invade and thrive in the gut[4]. This method creates large pools of mutants with unique barcodes for colonization studies but poses bioinformatic challenges, requiring many mice and high sequencing costs. In addition, genes of interest may be missing from the pool, requiring more mutants and mice for statistically significant data. Our study adopts a different approach, leveraging extensive knowledge of *S.* Typhimurium and *E. coli* central carbohydrate metabolism[32,33]. Recently, the wild-type isogenic standardized hybrid (WISH)-tag, a neutral genetic barcode for tracking strain abundance, was established for *S.* Typhimurium[34]. We created a pool of uniquely WISH-barcoded, rationally selected *S.* Typhimurium mutants in carbohydrate utilization to represent the metabolic capacity.

Colonization resistance, as highlighted by Campbell et al.[35], points out the pivotal role of the microbiome. This protection is achieved by a diverse microbiota depleting available nutrients and is enhanced by the presence of taxa closely related to the incoming pathogen. For instance, *E. coli* increases protection against *Salmonella* infection[36–38]. In many cases, the crucial factor for this protection is the metabolic resource overlap between the microbiota, key species, and the invading pathogen[39]. Therefore, understanding the metabolic systems used by *S.* Typhimurium in various microbiota contexts is essential. We studied the same *S.* Typhimurium mutant pool across five mouse models: germ-free C57BL/6J, streptomycin-pretreated C57BL/6J (specific pathogen-free, SPF) and 129S6/SvEvTac (SPF), low-complexity microbiome (LCM) C57BL/6J, and oligo mouse microbiota (OligoMM[12]) C57BL/6J mouse models (Fig. 1a). Unperturbed C57BL/6J (SPF) mice with an intact complex microbiome were not used due to the inability of *S.* Typhimurium to reproducibly colonize this particular model[40]. However, the LCM and OligoMM[12] models provide a balanced approach to studying pathogen colonization and interactions with the host microbiota, featuring partial colonization resistance that simulates natural infection kinetics. This study aimed to identify carbohydrate sources essential for *S.* Typhimurium colonization and examine their context dependency.

## Results

### Rationale for designing an *S.* Typhimurium carbohydrate mutant pool

A rationally designed mutant pool was constructed to study *S.* Typhimurium carbohydrate utilization across different mouse models, using *Salmonella enterica* subsp. 1 serovar Typhimurium SL1344[41]. We investigated 35 metabolic mutants using the WISH-barcoding approach[34], which allows amplicon sequencing for quantification of the strain abundance (Fig. 1b and Supplementary Data 1). The inoculum comprised four groups: control mutants with known colonization defects, seven wild-type strains to assess the evenness of distribution, a wild-type dilution series to establish the window of measurement, and 35 metabolic mutants deficient in carbohydrate-utilizing enzymes (Fig. 1c). The mutants, targeting carbohydrate-specific enzymes like kinases, dehydrogenases, or isomerases, provided insight into carbohydrate utilization without transporter specificity issues (Fig. 1d). This approach allowed us to systematically analyze the fitness of each

mutant in various gut environments, enabling the identification of context-independent and -dependent key carbohydrates essential for *S.* Typhimurium colonization. Four WISH-barcoded strains were included as controls. Firstly, a Δ*dcuA* Δ*dcuB* Δ*dcuC* triple mutant (abbreviated Δ*dcuABC*) and Δ*frdABCD* (abbreviated Δ*frd*), which encode the C4-dicarboxylate transporters and fumarate reductase, respectively, both crucial for fumarate respiration during gut-luminal growth in LCM mice[4] and during inflammation[42] (Fig. 1c). In addition, a Δ*hyb* mutant, encoding a hydrogenase important for utilizing hydrogen as an electron donor in LCM mice[21] and finally a SL1344 wild type was also included. To assess bottleneck severity, seven SL1344 wild-type strains were used to calculate the Shannon evenness score. A drop below 0.9 indicated a significant bottleneck, potentially leading to false positives; therefore, such samples were excluded[43]. A wild-type dilution series established the measurement window for each mouse and time point, typically displaying a difference of $10^{-3}$, ensuring accurate quantification within this limit of detection (see "Methods"; Fig. 1c). Each strain was WISH-barcoded for subsequent abundance quantification in fecal or cecum samples by amplicon sequencing, which was analyzed using the mBARq software[44].

We employed two primary infection protocols in our experiments (Fig. 1e, f). For SPF C57BL/6J and 129S6/SvEvTac mouse models, mice were pretreated with streptomycin to reduce resident microbiota and achieve consistent SL1344 infection[45,46]. We also used gnotobiotic mouse models with a C57BL/6J background: the low-complexity microbiota (LCM) model with 8 bacterial strains[21,47,48] and the oligo mouse microbiota model (OligoMM[12]) with 12 strains[36,49]. Germ-free mice, lacking any microbiome, were also included[50]. These non-SPF models were infected without antibiotic pretreatment (Fig. 1e, f). Due to the severity of SL1344 infection and population bottlenecks occurring during the late stage of infection[43], streptomycin-pretreated models were limited to a 2-day infection period. The same 2-day infection protocol was also used for germ-free mice (Fig. 1e). In contrast, infection of gnotobiotic animals could be extended to 4 days, as these models feature much less pronounced gut-luminal inflammation-associated bottlenecks. An inoculum size of $5 \times 10^7$ CFUs ensured each strain was represented by over 100,000 cells, minimizing stochastic effects.

### *S.* Typhimurium pool data was verified across all mouse models

The *S.* Typhimurium infection was monitored by plating feces and cecum content (cc) on selective MacConkey plates to quantify bacterial loads. As expected, gut colonization kinetics differed between models. In germ-free and streptomycin-pretreated mice, SL1344 reached $10^{10}$ CFUs per gram of stool within one day due to reduced microbiota or its absence (Supplementary Fig. 1a–c). In LCM and OligoMM[12] gnotobiotic models, colonization increased more slowly, saturating at day 3 and 4 post-infection, respectively (Supplementary Fig. 1d, e). The Shannon evenness scores for the seven SL1344 wild-type controls were close to 1 for the majority of samples in germ-free mice, streptomycin-pretreated mice, and the LCM mouse model (Supplementary Fig. 1f–i). The OligoMM[12] model showed more variation, with some samples falling below the 0.9 threshold by day 4 post-infection (Supplementary Fig. 1j). Samples with Shannon evenness scores below 0.9 were excluded from further analysis, ensuring high-quality fitness data for all models at all time points.

The WISH-barcoded SL1344 controls deficient in fumarate respiration (Δ*frd* and Δ*dcuABC*) showed 10 to 100-fold attenuation in all models compared to the wild type from day 1 post-infection (Supplementary Fig. 1k–o). Fumarate respiration can function independently of external fumarate by using monosaccharides or L-aspartate, making it the sole option for initial growth when external inorganic electron acceptors are absent[4,51,52]. The Hyb-deficient strain showed no attenuation and sometimes better growth in germ-free and streptomycin-pretreated models but exhibited 5 to 100-fold

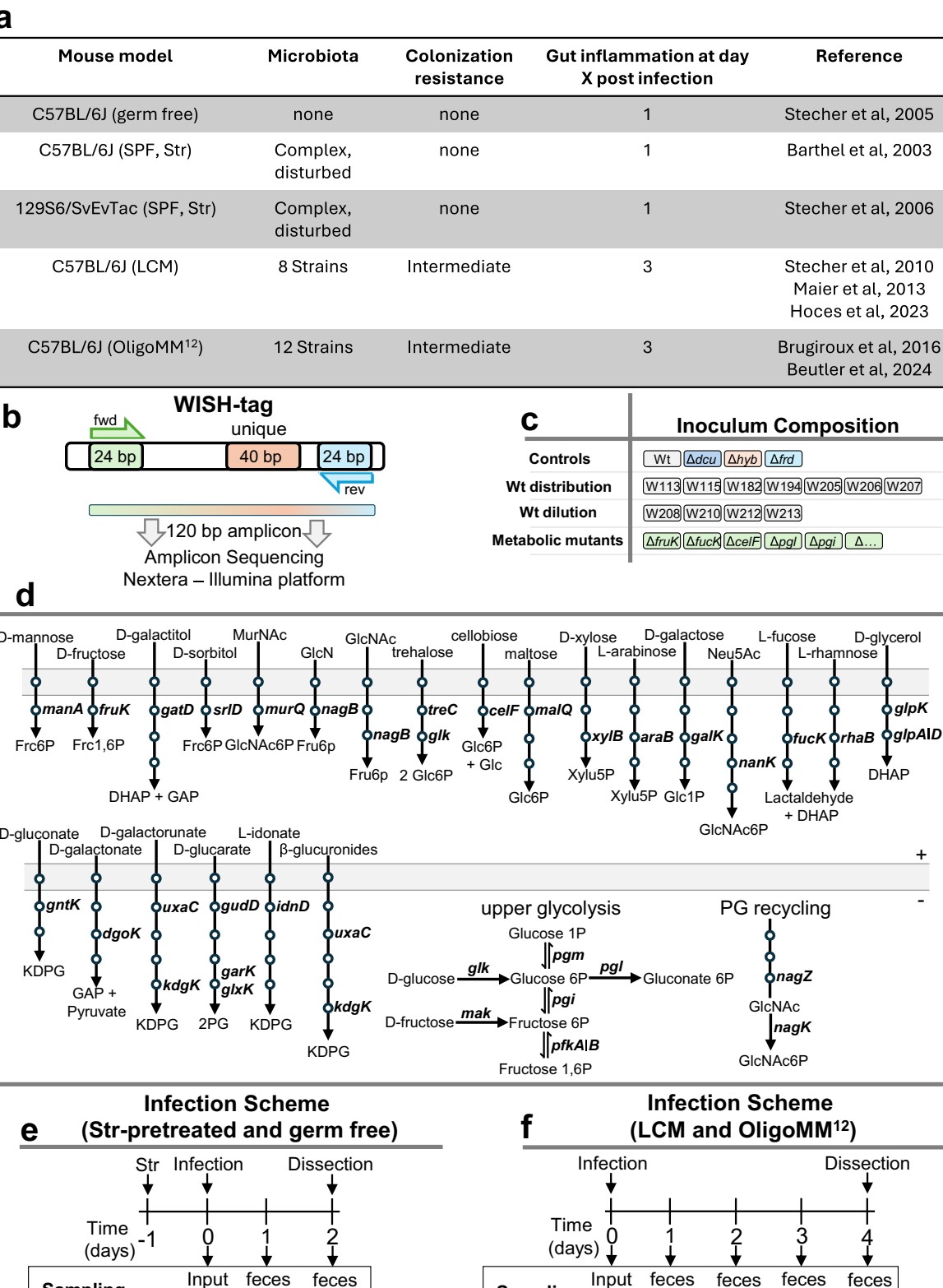

**Fig. 1 | Rational design of the WISH-barcoded *S.* Typhimurium mutant pool.**
**a** Overview of the five mouse models used in this study, showing microbiota composition, colonization resistance, and when gut inflammation typically occurs.
**b** Schematic representation of the wild-type isogenic standardized hybrid (WISH)-tag[34]. **c** The composition of the inoculum pool, as detailed in the main text, included a control wild type and mutants, seven wild-type strains for Shannon evenness score calculation, wild-type dilution standards to assess the limit of detection, and the carbohydrate mutants. **d** A schematic illustration shows the cytosolic genes selected for the mutant pool. Within each pathway, only the designated gene is highlighted, while each circle represents an enzymatic step in the degradation of the corresponding carbohydrate. Refer to Supplementary Data 12 for abbreviations. **e** Mouse infection protocol and sampling procedures for the germ-free (GF) and streptomycin (Str)-pretreated C57BL/6J and 129S6/SvEvTac mouse models.
**f** Mouse infection protocol and sampling procedures for the gnotobiotic low complex microbiota (LCM) and oligo mouse microbiota (OligoMM12) model. Abbreviations: cc, cecal contents.

attenuation in LCM and OligoMM[12] models (Supplementary Fig. 1k–m compared to Supplementary Fig. 1n, o). The LCM and OligoMM[12] model highlighted the importance of *hyb* for *S*. Typhimurium colonization, aligning with the expectation that a perturbed microbiota does not provide sufficient hydrogen to drive H₂/fumarate respiration[53–55]. These findings validate the experimental data, as previously published for LCM animals[4,21], and newly highlight the context-independent importance of fumarate respiration and the context-dependency of hydrogen utilization in *S*. Typhimurium colonization.

### D-fructose, D-galactose, and D-mannose utilization are crucial for colonization

Our SL1344 pool reveals both context-independent and context-dependent carbohydrate sources that are crucial for *S*. Typhimurium. In Supplementary Data 2–6, the competitive indices of all 35 metabolic mutants across five mouse models are listed and plotted as a heatmap in Supplementary Figs. 2 and 3. SL1344 mutants lacking *pgi* and *pgm* consistently fall below detection limits, underscoring the importance of glucose 1-phosphate, glucose 6-phosphate, and fructose 6-phosphate interconversion. The key enzyme of glycolysis, *pfkA*, which encodes 6-phosphofructokinase, was also important across all animal models; however, it was so highly attenuated that it consistently ranged close to the detection limit. In addition, the genes *fruK* (D-fructose), *galK* (D-galactose), and *manA* (D-mannose), which encode 1-phosphofructokinase, galactokinase, and mannose-6-phosphate isomerase respectively, are noteworthy. The *fruK*-deficient SL1344 mutant shows attenuation across all mouse models, with competitive index changes from 10 to 1000-fold (Fig. 2a–e). In streptomycin-pretreated models, the competitive index of *fruK* is near 1 on day 1, dropping 10-fold by day 2 post-infection (Fig. 2b, c), unlike other models where attenuation is seen on day 1 (Fig. 2a, d, e). Similarly, the *manA* mutant is attenuated in all models, with a significant drop below the limit of detection in almost all animals in the streptomycin-pretreated C57BL/6J model by day 1 post-infection (Fig. 2b). The *galK*-deficient mutant exhibits context-dependent attenuation only in germ-free and streptomycin-pretreated models, whereas *galK* mutants grew slightly better than the wild type in LCM and OligoMM[12] models. (Fig. 2a–c). These findings suggest a context-independent role for D-fructose and D-mannose utilization pathways in supporting *S*. Typhimurium growth in the murine gut, whereas the importance of D-galactose appears to be context-dependent in microbiota-perturbed mouse models.

### N-acetylglucosamine (*nagB*) and hexuronates (*kdgK*) are context-dependent nutrient sources

The *galK* and *hyb*-deficient SL1344 mutants accentuate context-dependency in carbohydrate and energy metabolism. Similarly, *nagB* and *kdgK* provide additional examples, as their attenuated fitness is observed only in a subset of mouse models. *nagB* encodes glucosamine-6-phosphate deaminase, important in *N*-acetylglucosamine degradation. A *nagB*-deficient SL1344 mutant shows a 5-fold attenuation in the germ-free model at day 2 post-infection, but a competitive index close to 1 in streptomycin-pretreated models. In LCM models, the *nagB* mutant shows a 50-fold attenuation, significantly more than in the OligoMM[12] model or any other model (Fig. 2f). This highlights and confirms the specific importance of NagB in the LCM model[40]. *kdgK* encodes 2-dehydro-3-deoxygluconokinase, involved in β-glucuronide (hexuronates) degradation. Post-antibiotic stress induces the inducible nitric oxide synthase (iNOS) in the cecal mucosa, leading to monosaccharide oxidation to sugar acids[56]. KdgK catalyzes the penultimate step in the degradation of D-fructuronate, D-mannonate, and D-glucuronate, preceding the Entner-Doudoroff pathway (Fig. 2g). The *kdgK*-deficient SL1344 mutant is attenuated only in streptomycin-pretreated and germ-free models, showing a wild-type-like competitive index in LCM and OligoMM[12] models (Fig. 2h). Previous studies suggested that

post-antibiotic stress creates a sugar acid and hexuronate-rich niche for *S*. Typhimurium colonization by oxidizing hexoses such as D-glucose and D-galactose[56], indicating a general mechanism that likely also oxidizes D-fructose and D-mannose. Future research should investigate the mechanisms that lead to an increase in hexuronates in the gut lumen of germ-free mice. Regardless, the Δ*kdgK* phenotypes are also consistent for Δ*uxaC* (D-galacturonate) mutants, which show consistently lower competitive index in streptomycin-pretreated models than in gnotobiotic models (Supplementary Fig. 4a). In contrast, Δ*dgoK* (D-galactonate) shows a 5- to 10-fold attenuation across the different mouse models (Supplementary Fig. 4b), indicating that this pattern is specific, as not every sugar acid displays it. Specifically, Δ*idnD* (L-idonate) shows no competitive disadvantage in all tested mouse models (Supplementary Fig. 4c). Thus, the Δ*nagB*, Δ*kdgK*, and Δ*uxaC* mutants further highlight the context-dependent nature of carbohydrate metabolism across different mouse models.

### D-fructose provides a niche important in intraspecies competition

To identify carbohydrates that are important for *S*. Typhimurium intraspecies competition, we analyzed the SL1344 mutant pool in the presence of an additional niche competitor, *S*. Typhimurium strain ATCC14028s. SL1344 can coexist with ATCC14028s during colonization due to differences in metabolic capacity, particularly in D-galactitol utilization, as previously shown[26]. For this reason, we aimed to investigate whether ATCC14028s alters the gut-luminal environment in a way that affects the fitness of specific carbohydrate mutants within our SL1344 mutant pool. Using the streptomycin-pretreated 129S6/SvEvTac mouse model, the SL1344 mutant pool was competed with ATCC14028s at ratios of 1:50 and 1:1000. Mice were infected with the same total *Salmonella* dose (5 × 10⁷ CFUs) as in previous experiments. Despite ATCC14028s's higher abundance in the inoculum, it did not alter SL1344 loads in fecal and cecal samples, consistent with previous findings[26]. SL1344 loads exceeded 10¹⁰ CFU per gram of feces by day 1 post-infection (Fig. 2i). Most samples had Shannon evenness scores above 0.9, allowing robust fitness evaluation (Fig. 2j). ATCC14028s only had a significant effect on three mutants: *pgl*, *pfkA*, and *fruK*, with *fruK* being roughly 100-fold attenuated at day 2 post-infection (Fig. 2k and Supplementary Fig. 5a, b). These findings indicate that carbohydrate competition is highly specific and potentially centered on D-fructose utilization for *Salmonella serovars*, with most SL1344 mutants showing no significant change in fitness in the presence of ATCC14028s. In Supplementary Data 7, the competitive index for all metabolic mutants in intraspecies competition is listed and plotted as a heatmap in Supplementary Fig. 5c–e.

### Hexoses provide the basis for successful colonization

Our previous experiments highlighted three key metabolic genes: *fruK*, *galK*, and *manA*, involved in the degradation of D-fructose, D-galactose, and D-mannose. A D-glucose-associated mutant was initially absent in the SL1344 pool due to the lack of an ideal D-glucose-specific cytosolic enzyme. Therefore, we investigated transporters to probe the importance of D-glucose. D-glucose can be transported by the primary transporter PtsG, as well as by several other transporters[32]. In the streptomycin-pretreated C57BL/6J model, the *ptsG*-deficient SL1344 mutant showed a 5-10-fold fitness defect from day 2 post-infection (Fig. 3a). However, this differed in the gnotobiotic LCM mouse model, where a *ptsG* mutant did not display a significant colonization defect during a 4-day infection period in a TnSeq colonization experiment[40]. Integrating *manA*, *fruK*, and *galK* mutations with *ptsG* deficiency aimed to further assess the importance of D-glucose utilization. Multimutants were constructed systematically, starting with the most attenuated mutant, *manA*, followed by *fruK*, *galK*, and finally *ptsG*.

To evaluate colonization capacity, SL1344 multimutants were tested in the streptomycin-pretreated C57BL/6J mouse model by

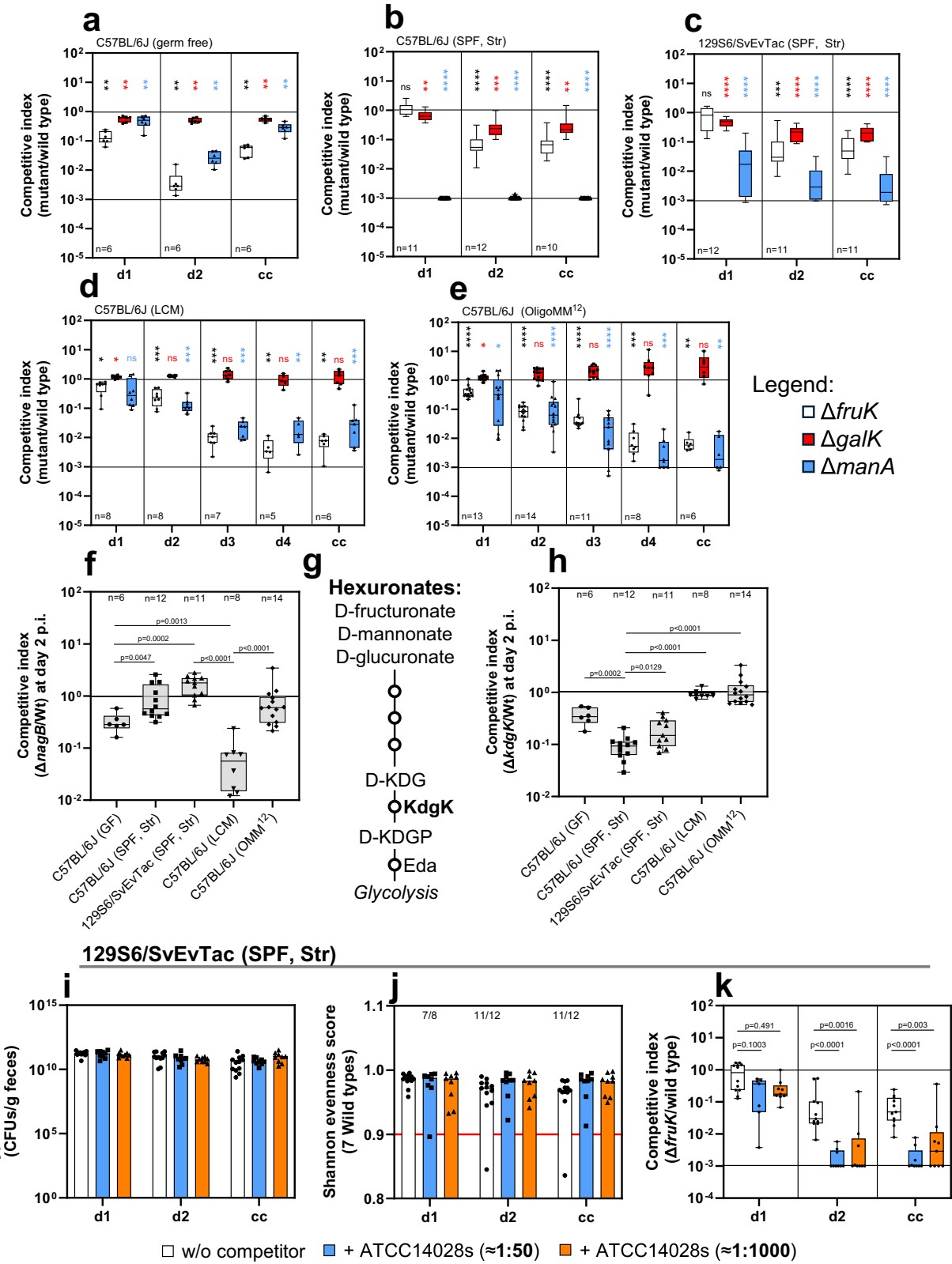

inoculating each group of mice with a single mutant (Fig. 3b). The SL1344 wild type reached ≈$10^{10}$ CFU/g feces by day 1 post-infection, maintaining this density until day 4. The *manA*-deficient strain grew significantly lower on day 1, staying roughly 100-fold lower than the wild type until the end of the experiment. The double (Δ*manA* Δ*fruK*) and triple (Δ*manA* Δ*fruK* Δ*galK*) mutants showed stable colonization over 4 days. However, while the quadruple mutant (Δ*manA* Δ*fruK*

Δ*galK* Δ*ptsG*) had similar loads to the triple mutant initially, densities collapsed by days 3-4 (Fig. 3b). Previous work showed gut microbiota regrowth by days 3-4 if *Salmonella* mutants fail to trigger gut inflammation in streptomycin pre-treated mice[15,57,58]. To test this, we used an avirulent isogenic mutant with disrupted type III secretion system-1 (T3SS, Δ*invG*) and T3SS-2 (Δ*ssaV*), which showed similar colonization kinetics to the quadruple mutant. However, we cannot determine

**Fig. 2 | S. Typhimurium mutants deficient in D-fructose, D-mannose, and D-galactose are attenuated.** Panels (**a**–**e**) show the competitive index for Δ*fruK* (D-fructose; white), Δ*galK* (D-galactose; red), and Δ*manA* (D-mannose; blue) across five mouse models, statistically compared to the SL1344 wild type. The upper black line indicates a wild-type competitive index of 1, and the lower line indicates the detection limit ($10^{-3}$). Sample sizes for each day post-infection are indicated in the subpanels. Panel (**f**) shows the competitive index for Δ*nagB* (N-acetylglucosamine), and panel (**h**) shows the competitive index for Δ*kdgK* (hexuronate) at 2 days post-infection across all models. Panel (**g**) depicts a schematic of hexuronate degradation, highlighting KdgK as a key enzyme. Panel (**i**) presents SL1344 CFUs/g feces for WISH-barcoded SL1344 pool in competition with ATCC14028s at 1:50 ($n = 9$) and 1:1000 ($n = 9$) ratios and in the absence of ATCC14028s ($n = 12$), from at least

independent experiments. Panel (**j**) shows Shannon evenness scores (SES) for the SL1344 mutant pool, with the number of datasets exceeding SES > 0.9 indicated. Panel (**k**) shows the Δ*fruK* competitive index in the presence of ATCC14028S at 1:50 ratios ($n = 7$ for day 1; $n = 9$ for day 2 and cecal content) and 1:1000 ratios ($n = 9$). It also includes data in the absence of ATCC14028S (day 1: $n = 12$; day 2 and cecal content: $n = 11$). Box-and-whisker plots (**a**–**f**, **h**, **k**) display the median, interquartile range (25th to 75th percentiles), minimum and maximum values, and individual data points. Bar plots (**i**, **j**) show medians with individual points. *P* values (Mann–Whitney U-test) are: **** ≙ $P < 0.0001$; *** ≙ $P < 0.0005$; ** ≙ $P < 0.005$; * ≙ $P < 0.05$; ns ≙ $P > 0.05$. The source data for panels (**a**–**i**) are provided in the Source Data file.

whether the avirulent quadruple mutant is displaced because of its failure to trigger inflammation or its inability to utilize key carbohydrate sources (Δ4 avirulent; Fig. 3b).

To assess the role of gut microbiota in the competitive fitness of carbohydrate utilization mutants, we tested the quadruple (Δ*manA* Δ*fruK* Δ*galK* Δ*ptsG*) mutant in the OligoMM[12] model against the SL1344 wild type. The single (Δ*manA*), double (Δ*manA* Δ*fruK*), and triple (Δ*manA* Δ*fruK* Δ*galK*) mutants were also tested. On day 1 post-infection, competitive indices were similar for all mutants (Fig. 3c). However, by days 3 and 4, distinct attenuation levels appeared, with a gradual decrease in fitness from single to quadruple mutants. By day 4, the quadruple mutant's bacterial loads in fecal and cecal samples were below the detection limit (Supplementary Fig. 6). This highlights the importance of D-glucose, D-fructose, D-mannose, and D-galactose utilization for successful S. Typhimurium colonization. Although the single mutant *galK* was initially not attenuated in the OligoMM[12] model (Fig. 2e), its addition to the multimutant caused a significant further fitness loss compared to the double mutant (compare Δ*manA* Δ*fruK* vs. Δ*manA* Δ*fruK* Δ*galK*, Fig. 3c). This was probably due to nutritional redundancy in the single *galK* mutant, suggesting that D-galactose is also a context-dependent nutrient source.

### The quadruple sugar mutant induces less inflammation than the wild type in the OligoMM[12] model

S. Typhimurium divides the labor of virulence expression into two phenotypically different subpopulations: virulent and avirulent, due to the high metabolic cost of virulence gene expression[59–61]. To further assess competition against the OligoMM[12] microbiota and the ability to cause inflammation, we performed single infections with wild-type SL1344, the quadruple mutant (Δ*manA* Δ*fruK* Δ*galK* Δ*ptsG*), or a triple mutant (Δ*fruK* Δ*galK* Δ*ptsG*), which retains a functional *manA* gene. The wild type reached ≈$10^9$ CFUs/g feces by day 3, while both mutants showed 100-fold lower colonization by days 2-3 post-infection (Fig. 3d). The triple mutant reached ≈$10^7$ CFUs/g feces faster than the quadruple mutant, but both strains reached this density by day 4. Unlike in the streptomycin-pretreated C57BL/6J model, the quadruple mutant stably colonized the gut without being displaced by regrowing microbiota, although at significantly lower levels than the wild type (Fig. 3d). These data show that while both mutants can stably colonize without wild-type competition, their population size is about 100-fold smaller. To evaluate inflammation, we measured lipocalin-2 in feces and analyzed cecum histopathology (Fig. 3e, f). Infection with both the wild type and the triple mutant resulted in high lipocalin-2 levels and severe enteropathy by day 4, indicating pronounced mucosal inflammation. Lower levels of inflammation were observed in mice infected with the quadruple mutant, with both lipocalin-2 and enteropathy (less edema) being significantly lower compared to the wild type (Fig. 3e–g). In conclusion, D-mannose (Δ*manA*), D-fructose (Δ*fruK*), D-galactose (Δ*galK*), and D-glucose (Δ*ptsG*) are essential nutrient sources for efficient colonization by S. Typhimurium and for inducing inflammation during the 4-day infection period in the OligoMM[12] model.

### S. Typhimurium exploits hexoses during colonization

The previous competition experiments focused on mutant fitness in classic 1:1 competitive experiments. In a different experimental setup, to determine whether a carbohydrate serves as an exploitable niche for S. Typhimurium wild type, we can supply the carbohydrate mutant in large excess compared to the wild type (with the wild type comprising 0.1% of the infection mixture). This allows us to see if the wild type can efficiently utilize the niche created by the mutant and reach similar levels within days, as previously shown for D-galactitol utilization[26]. This experimental change shifts the focus from the mutant to the wild type, excluding pleiotropic effects that can occur in metabolic mutants[62,63]. We performed these experiments with various carbohydrate utilization mutants in the streptomycin-pretreated C57BL/6J model. Two identical SL1344 wild-type strains with different antibiotic resistances maintained a stable gut density and ratio between each other (Fig. 4a). Using the *xylB* mutant (xylulokinase, D-xylose), which showed no change in competitive index (Supplementary Data 2–6), confirmed this (Fig. 4b). When there is no change in exploitable metabolic capacity, both strains will stay at the same density and ratio throughout the course of the experiment. The *manA*-deficient strain was displaced by the wild type by day 1 post-infection, consistent with its severe colonization defect in this mouse model (Fig. 4c). For *fruK* and *galK* mutants, the wild type caught up by day 2 post-infection (Fig. 4d, e). D-galactose provided a more favorable niche than D-fructose, as indicated by the wild type's faster catch-up kinetics on day 1 post-infection. The wild type also caught up with the *ptsG* mutant, though more slowly than with the *galK* and *fruK* mutant (Fig. 4f).

To further assess the roles of D-fructose and D-galactose utilization, we tested a *fruK galK* double mutant. The wild type not only caught up faster but displaced the double mutant within 2 days post-infection, a more pronounced phenotype than the single *fruK* or *galK* mutants (Fig. 4g compared to Fig. 4d, e). In conclusion, D-fructose, D-galactose, and D-glucose provide exploitable niches for S. Typhimurium. For D-mannose, it remains unclear if the displacement of the mutant or metabolic exploitation was decisive for the catch-up of the wild type. D-galactose is more favorable than D-fructose in the streptomycin-pretreated C57BL/6J model, even though the Δ*fruK* mutant's fitness was more attenuated than that of the Δ*galK* mutant (Fig. 2b). The role of D-glucose utilization is tougher to discern since the *ptsG* mutation only impairs one of the possible D-glucose transporters. However, inactivating PtsG creates a metabolic gap large enough to be exploited by the wild type.

### Free monosaccharides are abundant in cecum contents of mice

Our experiments show that genes associated with the utilization of D-mannose (*manA*) and D-fructose (*fruK*) are context-independent in our five mouse models of S. Typhimurium colonization. N-acetylglucosamine (*nagB*, GlcNAc), D-galactose (*galK*), and hexuronates (*kdgK*, β-glucuronide, *uxaC*, D-galacturonate) serve as context-dependent niches in certain mouse models. To determine carbohydrate availability in the cecum, we collected contents from gnotobiotic

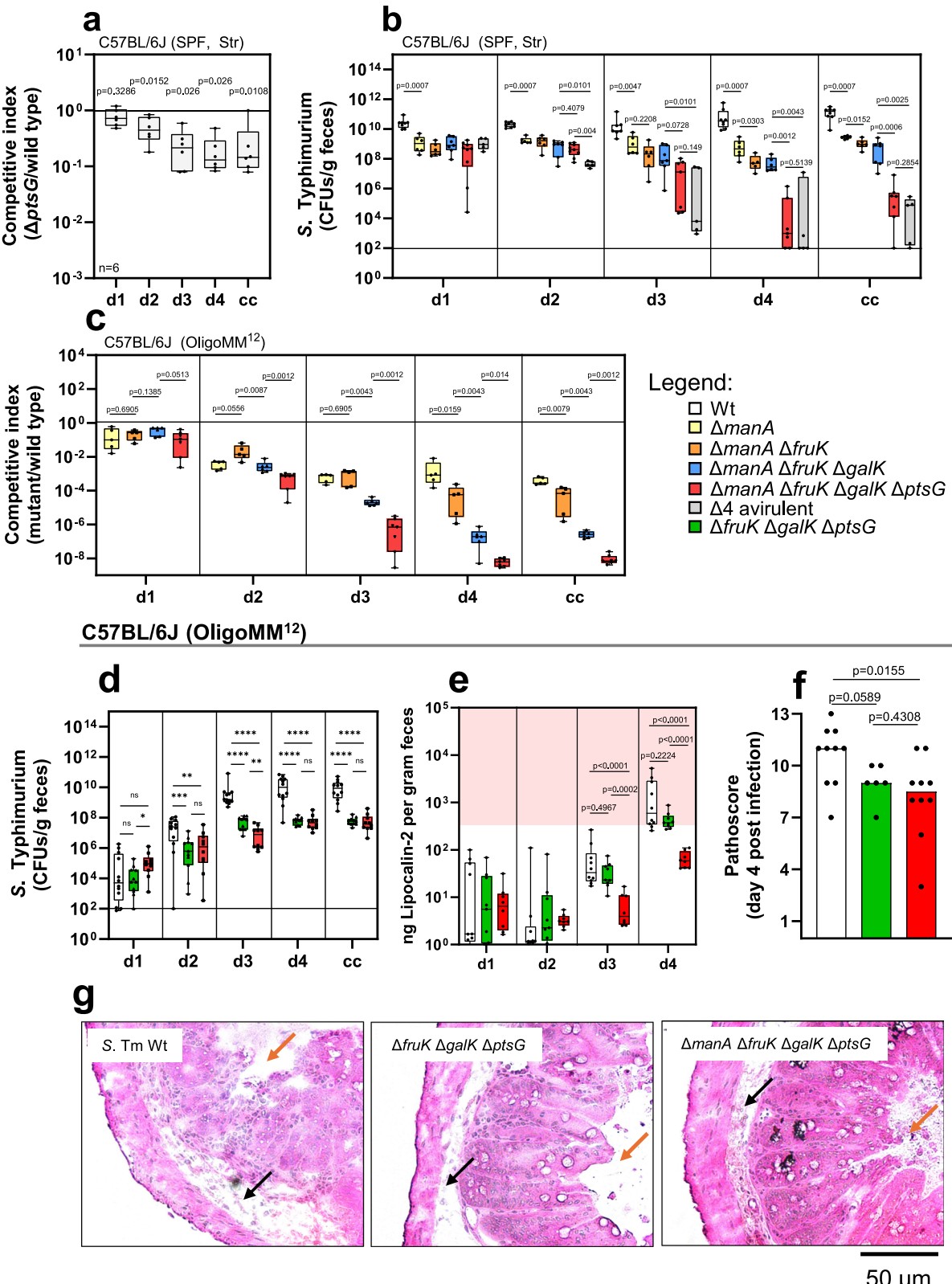

OligoMM[12] and germ-free mice and analyzed them using LC-MS. This was compared with previously published data from unperturbed C57BL/6J SPF mice and streptomycin-pretreated C57BL/6J SPF mice (Fig. 5a)[40]. The hexoses D-glucose, D-galactose, and D-mannose are consistently lower in the C57BL/6J SPF mice than in the germ-free, OligoMM[12], and streptomycin-pretreated models. Surprisingly, D-glucose is similarly abundant in the OligoMM[12] model as in germ-free

mice, indicating that the OligoMM[12] consortium does not specifically limit this essential nutrient (Supplementary Fig. 7a). A similar pattern is observed for amino sugars, which are generally lowest in C57BL/6J SPF mice and the OligoMM[12] model but highest in germ-free mice, reflecting competition for amino sugars and hexoses among the resident microbiota (Supplementary Fig. 7b). This pattern also applies to the pentose D-ribose. L-arabinose and D-xylose have the highest

**Fig. 3 | Hexoses provide the basis for *Salmonella* colonization. a** The competitive experiment was performed in streptomycin-pretreated C57BL/6J mice, lasting 4 days post-infection, comparing the SL1344 wild type with a *ptsG* mutant (mice *n* = 6, two independent experiments). **b** Single infections of streptomycin-pretreated C57BL/6J mice were performed with SL1344 wild type (*n* = 8), Δ*manA* (*n* = 6), Δ*manA* Δ*fruK* (*n* = 6), Δ*manA* Δ*fruK* Δ*galK* (*n* = 7), Δ*manA* Δ*fruK* Δ*galK* Δ*ptsG* quadruple mutant (*n* = 7), and an avirulent quadruple mutant (Δ*invG* Δ*ssaV*, *n* = 5). Bacterial loads (CFUs/g feces or cecal content (cc)) were recorded for days 1–4 post-infection. **c** Competitive experiments were conducted in C57BL/6J mice associated with OligoMM[12] microbiota, involving Δ*manA* (*n* = 5), Δ*manA* Δ*fruK* (*n* = 5), Δ*manA* Δ*fruK* Δ*galK* (*n* = 6), and Δ*manA* Δ*fruK* Δ*galK* Δ*ptsG* quadruple mutant (*n* = 7). These mutants were co-infected with the SL1344 wild type, with data collected from at least two independent experiments. **d** Single infections in OligoMM[12]-associated mice (wild type, *n* = 12; Δ*fruK* Δ*galK* Δ*ptsG*, *n* = 11; Δ*manA* Δ*fruK* Δ*galK* Δ*ptsG*, *n* = 10) recorded CFUs/g feces or cecal content. **e** Lipocalin−2

ELISA assessed gut inflammation in OligoMM[12]-associated mice (wild type, *n* = 10; Δ*fruK* Δ*galK* Δ*ptsG*, *n* = 9; Δ*manA* Δ*fruK* Δ*galK* Δ*ptsG*, *n* = 8). Inflammation was indicated at ≥500 ng lipocalin-2/g feces. **f** Histopathology scoring of cecum tissue at day 4 post-infection in OligoMM[12] mice (wild type, *n* = 10; Δ*fruK* Δ*galK* Δ*ptsG*, *n* = 7; Δ*manA* Δ*fruK* Δ*galK* Δ*ptsG*, *n* = 9). **g** Representative microscopy images of cecum tissue show gut edema (black arrow) and gut lumen (orange arrow) in OligoMM[12] mice infected with SL1344 wild type, Δ*fruK* Δ*galK* Δ*ptsG*, and Δ*manA* Δ*fruK* Δ*galK* Δ*ptsG* mutants. Panels (**a**–**e**) are presented as box-and-whiskers plots that display the median, interquartile range (25th to 75th percentiles), minimum and maximum values, and individual data points. Panel (**f**) are presented as bar plots, showing the median with all data points displayed. For panels (**a**–**f**), *P* values were calculated using the two-tailed Mann–Whitney *U*-test: **** ≙ *P* < 0.0001; *** ≙ *P* < 0.0005; ** ≙ *P* < 0.005; * ≙ *P* < 0.05; ns ≙ *P* > 0.05. The source data for panels (**a**–**g**) are provided in the Source Data file.

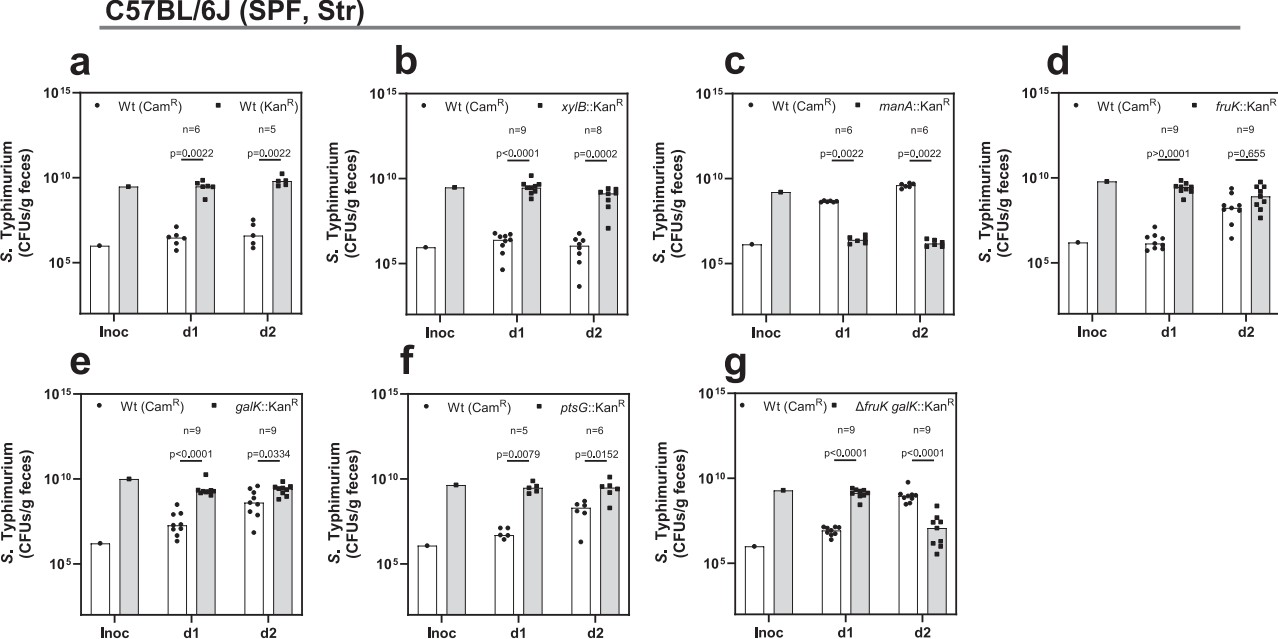

## C57BL/6J (SPF, Str)

**Fig. 4 | Hexoses provide an exploitable niche for *S.* Typhimurium.** For panels (**a**–**g**), the catch-up experiments were performed in the streptomycin-pretreated C57BL/6J model, with the carbohydrate utilization mutant in large excess and the SL1344 wild type constituting only 0.1% of the infection mix. The bacterial loads for each strain were determined by differential plating on MacConkey agar plates with kanamycin or chloramphenicol antibiotics. The individual bacterial loads are plotted in colony-forming units (CFUs) per gram of feces, including the inoculum

(adjusted to CFUs/g), with the strains indicated at the top of the respective subpanel for the 2-day infection period (number of mice *n* ≥ 6, at least two independent experiments). The bar plots show the median with all data points displayed. *P* values were calculated using the two-tailed Mann–Whitney *U*-test: **** ≙ *P* < 0.0001; *** ≙ *P* < 0.0005; ** ≙ *P* < 0.005; * ≙ *P* < 0.05; ns ≙ *P* > 0.05. The source data for panels (**a**–**g**) are provided in the Source Data file.

abundance in the streptomycin-pretreated C57BL/6J model. In principle, the microbiota can liberate L-arabinose and D-xylose from arabinan or xylan polymers without consuming all the freed monomers, as evidenced by the comparison between germ-free and OligoMM[12]-associated mice. Unperturbed SPF C57BL/6J mice have lower L-arabinose concentrations, suggesting that the microbiota is better able to limit these pentoses. However, it is unknown why streptomycin pretreatment drastically increases these particular pentoses (Supplementary Fig. 7c). Regarding deoxy sugars, L-rhamnose and L-fucose were previously shown to be of microbial origin[64]. These sugars can be utilized by some microbiota members for fermentation to produce the electron acceptor 1,2-propanediol[8]. Specifically, L-rhamnose is abundant in the OligoMM[12] and streptomycin-pretreated model compared to the other two, suggesting that L-rhamnose is produced but not efficiently consumed by the microbiota (Supplementary Fig. 7d). Hexuronates, such as D-glucuronic acid, D-guluronic acid, and D-mannuronic acid, are generally most abundant in germ-free and

streptomycin-pretreated C57BL/6J SPF mice, with the exception of D-galacturonic acid. The high concentration of D-glucuronic acid may explain the reduced fitness of the hexuronate-utilizing *S.* Typhimurium *kdgK* mutant in germ-free mice and streptomycin-pretreated C57BL/6J SPF mice (Supplementary Fig. 7e). In conclusion, D-glucose and D-galactose were found in the millimolar range, while D-mannose and *N*-acetylglucosamine were in the micromolar range. D-glucuronic acid, D-mannuronic acid, and D-guluronic acid concentrations varied substantially between models, being most abundant in germ-free and streptomycin-pretreated models (Fig. 5a and Supplementary Fig. 7e). The C57BL/6J SPF model consistently had lower free monosaccharide levels than the OligoMM[12] model, most likely due to consumption by the complex microbiome, reinforcing the essentiality of a complex microbiome for nutrient blocking.

Based on these considerations, we hypothesized that genes for utilizing critical carbohydrates fueling pathogen growth in the host's gut lumen should be universally present among *S. enterica* strains. To

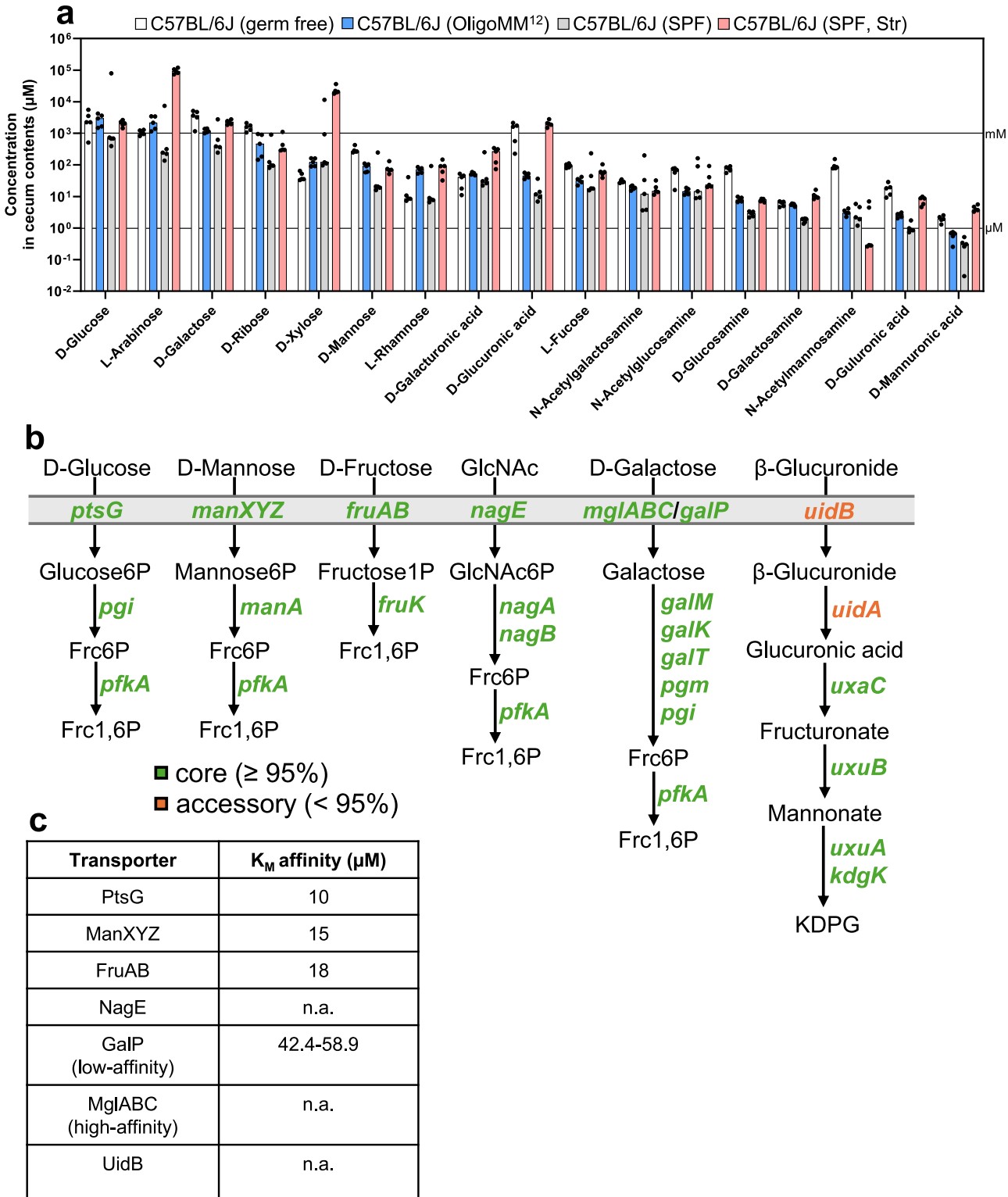

**Fig. 5 | Free monosaccharides are abundant in cecum contents of mice.**
**a** Absolute measurement of free monosaccharides in cecum contents of C57BL/6J mice associated with the OligoMM[12] microbiota (mice $n = 5$; blue) or germ free (mice $n = 5$, white) quantified by LC-MS. For comparison a previously published dataset of C57BL/6J with a complex specific pathogen-free (SPF) microbiota (mice $n = 5$, gray) and streptomycin-pretreated C57BL/6J SPF was included (mice $n = 5$, red)[40]. The black lines indicate mM and μM concentration, respectively. **b** A schematic representation of the D-glucose, D-mannose, D-fructose, *N*-

acetylglucosamine (GlcNAc), D-galactose, and β-glucuronide (hexuronate) degradation pathways. If a gene is present in at least 95% of all analyzed genomes, it is termed core (green). Less than 95%, it is termed accessory (orange). The full names of the abbreviations are listed in Supplementary Data 12. **c** Bacterial transporters with the respective $K_M$ affinity are shown in μM. PtsG and ManXYZ from *S.* Typhimurium[87]; FruAB from *P. aeruginosa*[88]; GalP from *E. coli*[89]. $K_M$ affinities for NagE, MglABC, and UidB are not available (n.a.). The source data for panels (**a**, **b**) are provided in the Source Data file.

understand the distribution of these metabolic systems within *Enterobacteriaceae*, we analyzed the presence of genes involved in carbohydrate utilization across four genera – non-typhoidal *Salmonella*, *Escherichia*, *Shigella*, and *Citrobacter* - which are known to colonize the mammalian gut and frequently cause disease. D-glucose, D-mannose, D-fructose, *N*-acetylglucosamine (GlcNAc), and D-galactose utilization genes are retained in over 95% of genomes analyzed (Fig. 5b). Only the transport and the initial step of β-glucuronide degradation are linked to the accessory genome, which are not found in *Salmonella* (Supplementary Data 8). To test the hypothesis that context-dependent carbohydrate utilization systems may tend to belong to the accessory genome, we extended this analysis to include D-xylose (*xyl* operon), hydrogen (*hyb* operon), L-idonate (*idn* operon), and D-galactonate (*dgo* operon). Hexuronate utilization are associated with the accessory genome in non-typhoidal *Salmonella*, *Escherichia*, *Shigella*, and *Citrobacter*, whereas D-xylose and hydrogen utilization is associated with the core genome (Supplementary Fig. 8 and Supplementary Data 8). Interestingly, D-xylose is relatively abundant in the cecum contents of mice (Fig. 5a); however, it does not appear to serve as a nutrient source for *S*. Typhimurium during colonization (Fig. 4b and Supplementary Figs. 2, 3). This finding highlights that *Enterobacteriaceae* are highly adapted to utilize these monosaccharides, which are particularly abundant in the gut of different mouse models, as reflected by the association of the degradation genes with the core genome.

With the exception of D-galactose, hexuronates, and D-xylose, D-glucose, D-mannose, D-fructose, and *N*-acetylglucosamine are transported by the phosphotransferase system (PTS)[65,66]. PTS transporters have high affinity for sugars in the low micromolar range, while GalP, a low-affinity transporter, has approximately 5-fold lower affinity (Fig. 5c). In contrast, human glucose transporters have affinities ranging between 0.2 and 17 mM[67]. This suggests that enteric bacteria have evolved to efficiently utilize monosaccharides in low micromolar concentrations, consuming what is not utilized by the host.

## Discussion

To assess the metabolic pathways that promote enteropathogen growth in the gut, we screened a mutant pool lacking key enzymes involved in carbohydrate utilization pathways of *S*. Typhimurium across five different mouse models. This approach establishes a fast and cost-effective method to determine the metabolic requirements of *S*. Typhimurium for colonizing different environments. This confirmed the critical role of $H_2$/fumarate respiration in the LCM gut[4,21] and emphasized the importance of fumarate respiration for *Enterobacteriaceae*[4,20,42,51,52,68] in a wide range of different mouse models. Here we further identified metabolic genes associated with the utilization of D-glucose (*ptsG*), D-mannose (*manA*), and D-fructose (*fruK*) to be context-independent, while D-galactose (*galK*), *N*-acetylglucosamine (*nagB*) and hexuronates (*kdgK*, and *uxaC*) are context-dependent for *S*. Typhimurium SL1344 colonization across various mouse models. Combining mutations of context-(in)dependent metabolic genes (Δ*manA* Δ*fruK* Δ*galK* Δ*ptsG*) in one SL1344 strain progressively attenuated gut-luminal growth, underscoring their significance as the basic metabolic requirements for successful *S*. Typhimurium colonization.

D-glucose, D-mannose, D-galactose, and D-fructose are prevalent in dietary plants, host glycans, and various other sources[69]. The monomer composition of different foods predominantly consists of D-glucose, reflecting its natural abundance[70–72]. Quantifying these carbohydrates in the cecal contents of gnotobiotic OligoMM[12] and germ-free mice revealed D-glucose and D-galactose in the millimolar range, while others, including *N*-acetylglucosamine and D-glucuronic acid, were in the micromolar range. These concentrations exceed bacterial transport affinities by one or two orders of magnitude (Fig. 5c). Interestingly, unperturbed SPF C57BL/6J mice had consistently lower concentrations of D-glucose, D-mannose, and D-galactose, compared

to the OligoMM[12] model, highlighting the importance of a complex microbiome for efficient nutrient blocking to provide colonization resistance. Even though monosaccharide concentrations are still considerable in SPF C57BL/6J mice, *S*. Typhimurium has substantial difficulty colonizing an unperturbed C57BL/6J SPF mouse model[40]. This suggests that either these monosaccharide concentrations are too low or that metabolic exploitation of monosaccharides is not the only key factor in preventing *S*. Typhimurium colonization. D-fructose was shown to be available in SPF C57BL/6 mice at concentrations that support *Enterococcus* colonization[73]. Furthermore, D-fructose seems to be an important nutrient niche for intraspecies competition between two *Salmonella* serovars. Glycan-degraders like *Bacteroides thetaiotaomicron* release monosaccharides from complex polysaccharides, making them accessible to other microbiota members or pathogens[74]. Free monosaccharide concentrations in germ-free mice often times exceeded those in the OligoMM[12] measurements, such as in the case of D-galactose and D-mannose (Fig. 5a). This suggests that a sterile host does not efficiently consume monosaccharides, highlighting the importance of the synergy between host and microbiota in limiting these nutrients.

Previous studies showed that pathogenic *E. coli* (EDL933) specifically utilizes D-galactose, hexuronates, D-mannose, and D-ribose, while commensal *E. coli* (MG1655) relies on D-gluconate and *N*-acetylneuraminic acid[75–77]. This emphasizes the fact that *S*. Typhimurium and *E. coli* have a highly similar metabolic resource overlap, underlining the observation that *E. coli* is a key competitor against pathogenic *S*. Typhimurium in the mammalian gut[38,39]. In conclusion, *E. coli* and *Salmonella* spp. prefer monosaccharides to support their growth in the mammalian gut lumen, as evidenced by the presence of the necessary metabolic enzymes in both genera (Fig. 5b). The availability of free monosaccharides in the mouse gut is sufficient for microbial utilization, highlighting the critical role of nutrient blocking by the microbiota in limiting pathogen invasion. Furthermore, phylogenetic analysis suggests that these metabolic systems are conserved across *Salmonella*, *Escherichia*, *Citrobacter*, and *Shigella*, indicating that the metabolic strategies presented here are common among *Enterobacteriaceae*.

In humans, two primary glucose transporters exist: the passive GLUT transporter and the sodium-dependent SGLT transporters, with a Michaelis-Menten constant ($K_M$) between 0.2 and 17 mM[67]. $K_M$ essentially refers to the amount of substrate required for an enzyme to operate at half its maximal velocity, wherein the substrate affinity can be markedly lower. This $K_M$ is an order of magnitude higher than that of bacterial hexose transporters, which can uptake monosaccharides potentially in the nanomolar range (Fig. 5c). Thus, even if free monosaccharide levels are lower than those in the OligoMM[12] model or SPF C57BL/6J mice (Fig. 5a), they can still support some level of *S*. Typhimurium colonization. In conclusion, we provided fitness information for 35 metabolic genes across five different mouse models to differentiate nutrient sources into context-independent and context-dependent. This information will guide the future design of probiotics and microbiota consortia to actively reduce these metabolites to limit non-typhoidal *Salmonella* infection.

## Methods

### Ethics statement

All animal experiments were performed in agreement with the guidelines of the Kantonales Veterinäramt Zürich under licenses ZH158/19, ZH108/22 and ZH109/2022, and the recommendations of the Federation of European Laboratory Animal Science Association (FELASA).

### Animals

We used male and female mice aged 7–12 weeks, randomly assigning animals of either sex to the experimental groups. The consideration of animal sex was not included in the study design. All mice were

maintained on the normal mouse chow (Kliba Nafag, 3537; autoclaved; per weight: 4.5% fat, 18.5% protein, 50% carbohydrates, 4.5% fiber). The mice originated from C57BL/6J or 129S6SvEv/Tac breeders initially obtained from Jackson Laboratories. Mice with a normal complex microbiota were specific pathogen-free (SPF) and bred under full barrier conditions in individually ventilated cage systems at the EPIC mouse facility of ETH Zurich (light/dark cycle 12:12 h, room temperature $21 \pm 1\,^{\circ}C$, humidity $50 \pm 10\%$). LCM mice[21,48] and OligoMM[12] mice[36] are ex-germ-free C57BL/6J animals that have been associated with either strains of the altered Schädler flora or a genome-guided selection of a representative group of 12 microbiota strains. They are bred in flexible film isolators under strict exclusion of microbial contamination at the isolator facility of the EPIC mouse facility (light/dark cycle 12:12 h, room temperature $21 \pm 1\,^{\circ}C$, humidity $50 \pm 10\%$). Germ-free C57BL/6J mice were bred in flexible film isolators under strict exclusion of microbial contamination at the isolator facility of the EPIC mouse facility, ETH Zurich (light/dark cycle 12:12 h, room temperature $21 \pm 1\,^{\circ}C$, humidity $50 \pm 10\%$). All studies were conducted in compliance with ethical and legal requirements and were reviewed and approved by the Kantonales Veterinäramt Zürich under licenses ZH158/19, ZH108/22, and ZH109/2022.

## Bacteria and culture conditions

All *Salmonella* strains are isogenic to *Salmonella* Typhimurium SB300, a re-isolate of SL1344. The names SB300 and SL1344 are used synonymously. They are listed in Supplementary Data 9. All plasmids used in this study are listed in Supplementary Data 10. All strains were routinely grown overnight at 37 °C in Lysogeny broth (LB) with agitation. Strains were stored at − 80 °C in peptone glycerol broth (2% w/v peptone, 5% v/v glycerol (99.7%)). Custom oligonucleotides were synthetized by Microsynth AG (Switzerland) and are listed in Supplementary Data 11.

## Homologous recombination by Lambda Red

Single-gene knockout strains were generated using the lambda-red single-step protocol[78]. Primers were designed with an approximately 40 bp overhanging region homologous to the genomic region of interest and a 20 bp binding region corresponding to the antibiotic resistance cassette (Supplementary Data 11). PCR amplification was performed using the plasmid pKD4 for kanamycin resistance or the pTWIST plasmids for WISH tags, which include an ampicillin resistance cassette. DreamTaq Master Mix (Thermo Fisher Scientific) was employed, followed by the digestion of the template DNA using FastDigest DpnI (Thermo Fisher Scientific). Subsequently, the PCR product was purified using the Qiagen DNA purification kit (Macherey-Nagel). SB300 with either the pKD46 or pSIM5 plasmid was cultured for 3 h at 30 °C until the early exponential phase, followed by induction with L-arabinose (10 mM, Sigma-Aldrich) or 42 °C for 20 min, respectively. The cells were washed in ice-cold glycerol (10% v/v) solution and concentrated 100-fold. Ultimately, the PCR product was transformed by electroshock (1.8 V at 5 ms), followed by regeneration in SOC (SOB pre-made mixture, Roth GmbH, and 50 mM D-glucose) medium for 2 h at 37 °C, ultimately plated on selective LB-agar plates. The success of the gene knockout was verified by gel electrophoresis and Sanger sequencing (Microsynth AG). Antibiotic resistance cassettes were eliminated via flippase FLP recombination[79].

## Homologous recombination by P22 phage transduction

P22 phage transduction was conducted by generating P22 phages containing the antibiotic resistance cassette inserted into the gene of interest from the defined single-gene deletion mutant collection of *S.* Typhimurium[80]. The single-gene knockout mutant was incubated overnight with the P22 phage generated from a wild-type SB300 background. The culture was treated with chloroform (1% v/v) for 15 min followed by centrifugation and subsequent sterile filtration

(0.44 μM pore size). The P22 phages were subsequently incubated with the recipient strain for 15 min at 37 °C and then plated on selective LB-agar plates. This was followed by two consecutive overnight streaks on selective LB-agar plates. Finally, the transduced clone was examined for P22 phage contamination using Evans Blue Uranine (EBU) LB-agar plates (0.4% w/v glucose, 0.001% w/v Evans Blue, 0.002% w/v Uranine). All mutations were verified by gel electrophoresis or Sanger sequencing (Microsynth AG), using the corresponding primers (Supplementary Data 11).

## WISH-barcoding of *S.* Typhimurium

WISH-barcodes were introduced, as previously described[34]. WISH-tags were amplified from pTWIST using DreamTaq Master Mix (Thermo Fisher Scientific) with WISH_int_fwd and WISH_int_rev primers (Supplementary Data 11) and integrated into *S.* Typhimurium SL1344 (strain SB300), using the λ-red system with pSIM5[78]. Integration was targeted at a fitness-neutral locus between the pseudogenes *malX* and *malY*, as previously described[81]. Correct integration was confirmed through colony PCR, and WISH-tags were validated by Sanger sequencing (*Microsynth AG*), using either the WISH_ver_fwd and WISH_ver_rev primers or the WISH_seq_fwd and WISH_seq_rev primers, respectively (Supplementary Data 11). Subsequently, P22 phage lysates were prepared from these generated strains to transduce the WISH-tag into SB300 wild types, controls, and carbohydrate utilization mutants.

## Preparation of the *S.* Typhimurium mutant pool

An individual clone was inoculated in selective LB media and grown overnight at 37 °C, containing carbenicillin (100 μg/ml) and kanamycin (50 μg/ml), or chloramphenicol (7.5 μg/ml) to select for the WISH-barcode and the respective mutation. The overnight culture was inoculated into a subculture (5% v/v) and grown for 4 h at 37 °C to the late exponential phase in LB media without antibiotics. At this stage, the strains were pooled in equal volumes, followed by centrifugation ($4500 \times g$, 4 °C, 15 min). The cell pellets were resuspended in peptone glycerol media to 10% of the original volume and aliquoted in 100 μl volumes in cryo tubes. The pools were stored at − 80 °C. This method allows for rapid utilization of the mutant pools, as 20 μl of the SL1344 mutant pool stock is subcultured in LB media for 4 h at 37 °C prior to infection. To investigate the colonization defects of metabolic mutants in various mouse models, we used aliquots from the same SL1344 mutant pool stocks throughout this study.

## Mouse colonization experiments

For the SL1344 mutant pool experiments, 7 to 12-week-old mice were inoculated with cells prepared from a 4 h subculture in LB medium sourced from the pre-mixed cryo stocks. For single and competitive infection experiments, 7 to 12-week-old mice were inoculated with *S.* Typhimurium cells prepared from a culture grown overnight in selective LB and subcultured for 4 h at 37 °C in LB media. The overnight culture was then used to inoculate (5% v/v) a subculture for 4 h in LB media without antibiotics at 37 °C. In both cases, bacteria were washed once with 1xPBS, and mice were infected with bacteria by gavage ($5 \times 10^7$ CFU in 50 μl). For the competitive catch-up experiments, the wild-type SL1344 strain (chloramphenicol-resistant) was diluted 1000-fold and then mixed with either another wild type or a carbohydrate mutant (kanamycin-resistant) in a 50 μl inoculum mix in equivolume. At the end of the experiments, either day 2 or 4 post-infection, animals were euthanized by $CO_2$ asphyxiation. Fresh fecal pellets and whole cecal content were sampled and homogenized in 1xPBS (500 μl + metal bead) using a TissueLyser device (Qiagen) before plating to determine the total bacterial population size.

## Sample preparation for the WISH barcode counting

After homogenization, fecal and cecal *S.* Typhimurium cells (125 μl) were enriched in 1 ml LB medium with 100 μg/ml carbenicillin (Carl

Roth GmbH) for 4 h at 37 °C to select for WISH-barcoded strains. Bacterial cells were pelleted, the supernatant was discarded and then stored at − 20 °C. DNA extraction from thawed pellets was performed using commercial kits (Qiagen Mini DNA kit) according to the manufacturer's instructions. For PCR amplification of the WISH-barcodes, 2 μl of the isolated genomic DNA sample and 0.5 μM of each primer (WISH_Illumina_fwd and WISH_Illumina_rev, see Supplementary Data 11) were used in a DreamTaq MasterMix (Thermo Fisher Scientific). The reaction was conducted with the following cycling program: initial denaturation step at (1) 95 °C for 3 min followed by (2) 95 °C for 30 sec, (3) 55 °C for 30 sec, (4) 72 °C for 20 sec, (5) repeat steps (2–4) for 25 cycles, and a terminal extension step at (6) 72 °C for 10 min. PCR products were column purified. We indexed the PCR products for Illumina sequencing by performing a second PCR with nested unique dual index primers using the following program: (1) 95 °C for 3 min, (2) 95 °C for 30 s, (3) 55 °C for 30 s, (4) 72 °C for 20 sec, (5) repeat steps (2–4) for 10 cycles, (6) 72 °C for 3 min. Afterward, we assessed the indexed PCR product using gel electrophoresis (1% w/v agarose, 1xTAE buffer), pooled the indexed samples according to band intensity, and subsequently purified the library via AMPure bead cleanup (Beckman Colter) before proceeding to Illumina sequencing. Amplicon sequencing was performed by BMKGENE (Münster, Germany). BMKGENE was tasked with sequencing each sample at a 1 G output on the NGS Novaseq platform, utilizing a 150 bp paired-end reads program. Subsequently, the reads were demultiplexed and grouped by WISH-tags using mBARq software[44]. Misreads or mutations of up to five bases were assigned to the closest correct WISH-tag sequence. The WISH barcode counts for each mouse in every experiment are available in Supplementary Data 2–7. These counts were used to calculate the competitive fitness and Shannon evenness score (7 wild types). WISH counts with less than or equal to 10 was excluded from further analysis and defined as the detection limit, as previously established[34]. The Shannon evenness score of the inoculum, excluding the SL1344 wild-type dilution series, was 0.99 after 4 h of enrichment in selective LB media, indicating no general growth defects in any strain within the pool (Supplementary Fig. 9a). In the streptomycin-pretreated C57BL/6J and the gnotobiotic C57BL/6J (OligoMM[12]) mouse model, wild-type dilutions at $10^{-3}$ were consistently detectable (Supplementary Fig. 9b and c). This trend was consistent across all tested models. Based on the minimum number of WISH counts (≥10) and the dilution standard, the competitive index of mutants that were below the limit of detection were conservatively set to a competitive index of $10^{-3}$.

## Calculation of the competitive index

To calculate the competitive index for the mutant pool, the values were determined by dividing the number of observed barcode reads at a specific time point (day 1 post-infection to day 4 post-infection or in cecum content) by the number of barcode reads observed in the inoculum. For the calculation of the competitive index, the individual strain fitness of each WISH-barcoded mutant were divided by the mean fitness value of the 7 WISH-barcoded wild-type S. Typhimurium control strains. To calculate the statistical significance, the metabolic mutants were compared to the SL1344 wild type in the control group. The raw data and calculations are available in the Supplementary Data 2–7 for every mouse model. The fitness of the competitive infections with SL1344 wild type against the carbohydrate utilization single- and multimutants was calculated by dividing the colony-forming units (CFUs) per gram of feces of the wild type by those of the mutants.

## Sample preparation for free monosaccharide quantification by LC-MS

Cecum contents were collected in the morning, i.e., the 2nd hour of the light phase in the mouse room, suspended in 1xPBS (1:1 ratio) and homogenized (without metal beads) using a TissueLyser (Qiagen) at 25 Hz for 2 min. This was followed by centrifugation for

5 min at 20,000 × g at 4 °C to pellet cells and particulate matter. Cell-free supernatants were collected and further centrifuged for 40 min at 20,000 × g to remove any remaining suspended particulates. Clear supernatants were then transferred to clean microfuge tubes and stored at − 80 °C until thawed for LC-MS processing. The sample size ($n$) per mouse model was $n = 5$ biological replicates.

## Free monosaccharide quantification by LC-MS

Following a previously published protocol[82], samples containing free monosaccharides (25 μL) were derivatized with 75 μL of 0.1 M 1-phenyl-3-methyl-5-pyrazolone (PMP) in 2:1 methanol:ddH$_2$O with 0.4 % ammonium hydroxide for 100 minutes at 70 °C. In addition, each sample was spiked with an internal standard of 10 μM $^{13}$C$_6$-glucose, $^{13}$C$_6$-galactose, and $^{13}$C$_6$-mannose (mass 186 Da). For quantification, we derivatized a serial dilution of a standard mix containing D-galacturonic acid, D-glucuronic acid, D-mannuronic acid, D-guluronic acid, D-xylose, L-arabinose, D-glucosamine, L-fucose, D-glucose, D-galactose, D-mannose, N-acetyl-D-glucosamine, N-acetyl-D-galactosamine, N-acetyl-D-mannosamine, D-ribose, L-rhamnose and D-galactosamine. Samples and standards were derivatized by incubation at 70 °C for 100 min. After derivatization, samples were neutralized with HCl, followed by chloroform extraction to remove underivatized PMP, as previously described.

Following[83], PMP-derivatives were measured on a SCIEX qTRAP5500 and an Agilent 1290 Infinity II LC system equipped with a Waters CORTECS UPLC C18 Column, 90 Å, 1.6 μm, 2.1 mm × 50 mm reversed phase column with guard column controlled with SCIEX Analyst v. 1.7.3. The mobile phase consisted of buffer A (10 mM NH$_4$Formate in ddH$_2$O, 0.1% formic acid) and buffer B (100% acetonitrile, 0.1% formic acid). PMP-derivatives were separated with an initial isocratic flow of 15% buffer B for 2 min, followed by a gradient from 15% to 20% buffer B over 5 minutes at a constant flow rate of 0.5 ml/min and a column temperature of 50 °C. The ESI source settings were 625 °C, with curtain gas set to 30 (arbitrary units), collision gas to medium, ion spray voltage 5500 (arbitrary units), temperature to 625 °C, and Ion source Gas 1 & 2 to 90 (arbitrary units). PMP-derivatives were measured by multiple reaction monitoring (MRM) in positive mode with previously optimized transitions and collision energies. For example, a D-glucose derivative has a Q1 mass of 511 and was fragmented with a collision energy of 35 V to yield the quantifier ion of 175 Da and the diagnostic fragment of 217 Da. Different PMP derivatives were identified by their mass and retention in comparison to known standards. Peak areas were integrated using Skyline 23.1.0.455 and concentrations were quantified using a custom Python v. 3.9. script. In short, technical variations in sample processing were normalized by the amount of internal standard in each sample. Peak areas of the 175 Da fragment were used for quantification using a linear fit to an external standard curve in triplicates with known concentrations ranging from 100 pM to 10 μM.

## Haematoxylin and eosin staining of tissue

Haematoxylin and eosin (HE) staining of cryo-embedded tissues (OCT, Sysmex Digitana) and subsequent pathoscoring for granulocyte infiltration were performed as previously described[45]. The Source Data file includes the individual pathological scores for four subgroups: submucosal edema, epithelial disruption, goblet cell depletion, and neutrophil infiltration. Each score is accompanied by a brief description detailing the different levels of intensity.

## Lipocalin-2 analysis of feces samples

Lipocalin-2 was detected in fecal samples homogenized in 500 μl sterile 1xPBS using an ELISA assay (DuoSet Lipocalin ELISA kit, DY1857, R&D Systems, Minneapolis, MN, USA). Fecal pellets were diluted 1:20, 1:400, or left undiluted, and concentrations were determined using Four-Parametric Logistic Regression curve fitting.

## Distribution of monosaccharide utilization genes in enteric bacteria

The distribution of genes involved in monosaccharide utilization were based on orthologous gene clusters generated from a dataset of 1158 *Enterobacteriaceae* genomes[84]. In short, Cherrak et al., created orthologous gene clusters at different sequence identity thresholds using PIRATE[85]. Gene clusters with relevant functional annotations (Fig. 5b and Supplementary Fig. 8) were extracted, and distribution patterns were assessed using Python 3.7.6, initially across four *Enterobacteriaceae* genera: *Citrobacter, Escherichia, Shigella,* and non-typhoidal *Salmonella* (Supplementary Data 8). Each genus was randomly subsampled to 65 genomes to ensure even distribution. Gene clusters were classified as core if present in at least 95% of the selected genomes. All other gene clusters were classified as accessory.

## Statistical analysis

No statistical methods were used to predetermine sample sizes. For all mouse experiments, sample sizes of 5 or greater were used, consistent with those reported in previous publications[86]. Data collection and analysis were not performed blind to the conditions of the experiments. Only animals with a Shannon evenness score below 0.9, as previously determined as the cutoff[43], were excluded from the WISH-pool analyses; otherwise, no animals were excluded. The excluded animals and their corresponding Shannon Evenness scores are highlighted in Supplementary Data 2–7. Where applicable, the two-tailed Mann-Whitney U test was employed to assess statistical significance, as specified in the figure legends. Statistical analyses were performed using GraphPad Prism 10 for Windows. $P$ values were grouped as follows: **** $\triangleq P < 0.0001$; *** $\triangleq P < 0.0005$; ** $\triangleq P < 0.005$; * $\triangleq P < 0.05$; ns $\triangleq P > 0.05$.

## Reporting summary

Further information on research design is available in the Nature Portfolio Reporting Summary linked to this article.

## Data availability

The amplicon sequencing data from the WISH-barcoded *S.* Typhimurium experiments generated in this study are available in the European Nucleotide Archive (ENA) under accession number PRJEB83965. Source data are provided in this paper.

## Code availability

The code used for WISH-barcode counting is available on GitHub https://github.com/MicrobiologyETHZ/mbarq [44].

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

## Acknowledgements

We would like to acknowledge and thank the staff at the ETH animal facilities (EPIC and RCHCI; especially Manuela Graf, Katharina Holzinger, Dennis Mollenhauer, Sven Nowok, Samuel Boateng, and Dominik Bacovcin), and extend many thanks to members of the Hardt, Sunagawa, Vorholt, and Slack labs, as well as the NCCR Microbiomes, for their helpful comments and discussions. Many thanks to Yassine Cherrak, Leanid Laganenka, and Gottfried Unden for providing helpful comments. This work has been funded by grants from the Swiss National Science Foundation (310030_192567, 10.001.588 and NCCR Microbiomes grant 51NF40_180575) to W.-D.H. C.S is supported by the German Research Foundation (SCHU 3606/1-1). This work has been further funded by grants from the Swiss National Science Foundation (grant 310030_19256) attributed to N.N. and C.v.M. and NCCR Microbiomes grant 51NF40_180575 to C.v.M.

## Author contributions

C.S. and W.-D.H. conceived and designed the experiments. C.S. performed the in vivo experiments. B.D.N., A.S., J.N., and U.S. collected samples and analyzed monosaccharides by LC-MS. A.S. and L.F. provided the bioinformatics pipeline for WISH analysis. B.B.J.D. and Y.S. provided support for WISH-barcoding. N.N. and C.v.M. performed the bioinformatic analysis for the presence of metabolic systems. C.S. wrote the manuscript with contributions from all authors.

## Competing interests

The authors declare no competing interests.
