## [Transparent Peer Review file · Nature Communications]

Monosaccharides drive *Salmonella* gut colonization in a context-dependent or -independent manner

Corresponding Author: Professor Wolf-Dietrich Hardt

Version 0:

Reviewer comments:

Reviewer #1

(Remarks to the Author)

In this paper, the authors use a multiplexed barcoding approach to assess the fitness of several metabolic salmonella mutants in various mouse models with altered microbiota. The authors find that some mutants have fitness defects across all mouse models, while others are only apparent in some mouse models. Overall this paper is a very deep analysis of carbohydrate utilization requirements for Salmonella and will be of interest to those who study Salmonella or carbohydrate utilization of other enteric pathogens. In my opinion no further experiments are required for publication, but there is an important experimental group missing (unperturbed C57BL/6J mice and some parts of the paper are difficult to read, so I have a few suggestions:

1. Experiments in this paper are well controlled and I do not believe any additional experiments are necessary for publication. One suggestion – since the authors already have access to this data – is to plot the evenness (of WT) as the X axis and mutant fitness on the Y axis. I am very curious if the fitness defects of individual mutants generally correlates with the severity of the infection bottleneck.
2. One caveat of this study is that unperturbed C57BL/6J mice are not included in competition experiments. I know that including streptomycin is helpful to improve colonization, but I believe that the “no step” model is used by others in the field. It would strengthen the paper (but is not necessary for publication) to test whether even a few mutants are defective for colonization in WT unperturbed C57BL/6J mice. In the absence of additional experiments, please directly reference this caveat
3. Please include a summary figure that provides the competitive indices for all mutants in this study. I appreciate how each figure tells a smaller story, but I find myself flipping back and forth quite a bit as I am reading this paper, and reading the raw count data in the supplementary tables is difficult
4. Line 18 - Including the WISH acronym as the 3rd sentence in the abstract without explaining what it is may be difficult to follow for readers who are unfamiliar with the technique
5. Line “five different mouse models” – a brief clarification on what these are could help with understanding (e.g., they could be different strains of mice, different routes of infection, etc.)
6. Line 57 – Lack of insertions within some genes is certainly an issue for some TnSeq libraries, but I do not think that target genes being “not effectively deactivated” is generally believed in the field. Consider omitting this point or providing a reference to show ineffective inactivation of random Tn insertions.
7. Line 60 – WISH may not be known to others in the field at this point in the manuscript. Consider elaborating
8. Line 60 – While I appreciate that rational design of mutants is a good strategy, it seems a bit odd to frame these experiments as overcoming limitations to TnSeq. TnSeq and barcoded mutant screens are not very comparable since one a

screen (forward genetics) and the other is not (reverse genetics).

9.Line 80 – the text mentions qPCR and amplicon sequencing, but Figure 4B shows only amplicon sequencing. I'm a bit confused as to how qPCR can be used here. Would the authors perform 35 qPCR reactions per sample?

10.Line 91 – is the hyb mutant a positive (fitness defect) or negative (no fitness defect) control

11.Line 106 – I am surprised that germ-free mice have severe bottlenecks. Is there a reference for this? Ref 41 does not appear to have germ free data. It also seems like Figure S1 shows less severe bottlenecks in GF mice at d2 (s1f) than OligoMM12 mice at d2 (s1j)

12.Figure S1 – please indicate what “cc” is in the figure itself

13.Line 138 – please include a figure, or at least a supplementary figure, that shows these results. It is difficult to check raw count tables

14.Line 182 – provide figure for “did not alter SL1344 loads”, since Figure 2i appears to only show general Salmonella loads. Please also specify here whether these experiments were done at the same total Salmonella dose as others

15.I would like the authors to elaborate more on what the rationale for using ATCC14028S is in the paragraph starting at line 178. Since experiments thus far use a mixture of strains, it seems that we already know that these mutants are important for “intraspecies competition”, as all experiments so far have been competitive infections with Salmonella

16.Line 197 – the authors say that the ptsG mutant did not have a defect in the LCM model and cite ref 52, but could they elaborate on how comparable these different experiments are? They may not be comparable if they were very different experimental designs. I cannot evaluate since ref 52 does not appear to be accessible online

17.Line 211. I don't follow this logic. The quadruple mutant has a strong colonization defect, so this fact alone would suggest it causes less inflammation. I don't think that adding the T3SS mutations onto the quadruple mutant shows that the quadruple mutant does not trigger inflammation. It also seems like 3d does show that inflammation is occurring the quadruple mutant in the same model, although less than WT

18.Please add arrows in 3g

19.Line 245 – is this true? Presumably each mutant is only 1/~50 in the whole pool, so isn't each mutant only about 2% of the population?

20.Line 273 – I feel like this is the most important sentence in the results, so I suggest that it is restated in the abstract and intro.

21.Figure 5 – the authors should mention the caveat that the SPF mice were not treated with streptomycin, which makes these results not comparable to other experiments in the paper

22.If possible, please include at least one example of where the carbohydrate concentrations (as quantified in figure 5) correlates with the fitness of the mutant across those 3 mouse models

23.Line 301 – while I definitely appreciate the phylogenetics, this section seems a little underbaked compared to the rest of the paper, especially for being at the end. For example, are these genes present in other organisms that do not live in the gut? Not a requirement to publish, but I would encourage the authors to consider changing its location in the manuscript, deepening the analysis, keeping it only as discussion, or omitting it entirely.

24.Figure legends – please include how many replicates are used whenever boxplots are displayed (e.g. figure 2a)

Reviewer #2

(Remarks to the Author)

This study examines the carbohydrate sources required for Salmonella to colonize animals. It uses several murine models, including those with complex but disrupted intestinal microbiota, and those with defined microbiota, as well as germfree mice. Using uniquely tagged strains in competition, the authors assess survival by competition assays to identify the important pathways for carbohydrate utilization. They present evidence that D-glucose, D-mannose, D-fructose, and D-galactose are used in all models tested, with other carbohydrates being important only under specific conditions, suggesting differences in carbohydrate utilization and productions among them. The work described was well conceived and carefully performed to reduce bias. It has resulted in new and interesting information. Some of the conclusions drawn, however, are not well explained or supported. Specific comments:

1. Line 82: The dilutions made using the wild-type strain are not well described.

2. Line 150: The galK mutant was found to be deficient in only germfree and streptomycin-treated mice. How can this be

explained? One hypothesis would be that some specific constituents of the microbiota produce galactose, but this does not appear to be true (Fig. 5a). What model can be proposed to explain this finding?

3. Line 159: The nagB mutant has a different, but also inconsistent phenotype: deficient in germfree and low-complexity microbiome mice, but not in those treated with streptomycin. Explanations for this?

4. Line 167: The phenotype of the kdgK mutant appears to be similar to that of the galK mutant. In this case, "post-antibiotic stress" is cited as a possible cause. In germfree mice, however, no antibiotic has been used. How would this apply to germfree animals?

5. Line 209: An argument is made that the manA, fruK, galK, ptsG mutant is incapable of triggering inflammation because its colonization is indistinguishable from that of an isogenic mutant with disruptions of the type III secretions systems required to produce intestinal inflammation (Fig. 3B). It seems equally plausible that this quadruple mutant can induce inflammation but fails to survive because of its inability to utilize important carbohydrates. Inflammation could have been directly tested, using the methods of Fig. 3e-g, but the two experiments used different mouse models, and the avirulent mutant was not tested for its inflammatory effects.

6. Line 237: The inflammatory markers (Fig. 3e-f) did not differ between the wild type and the fruK, galK,ptsG mutant. Only the additional loss of manA produced a difference. Does this implicate D-mannose specifically as the important carbohydrate? The manA mutant alone is deficient in both colonization and competitive index (Fig. 3b-c).

7. Line 301: The genes shown in this study to be important for Salmonella colonization are demonstrated also to be widely present in members of the Enterobacteriaceae. This evidence is used to argue for their broader importance for survival in animals. If correct, one might also expect the converse to be true: pathways not important for colonization would be less widely shared. Is this the case? Several genes used in this study could be tested, including kdgK, xylB, hyb, idnD, and possibly uxaC.

8. Comparing the results of Fig. 2e and Fig. 3c, the galK mutation produces no effect singly but demonstrates one only when combined with manA, fruK, and ptsG mutations. Does this indicate that galactose is not a preferred energy source for Salmonella in the intestine and is used only when other more favorable sources are not available?

Reviewer #3

(Remarks to the Author)

Version 1:

Reviewer comments:

Reviewer #1

(Remarks to the Author)

All of my concerns have been addressed, and I congratulate the authors on this very interesting paper.

Reviewer #2

(Remarks to the Author)

Thank you to the authors for their thoughtful comments. They have sufficiently addressed the concerns of this reviewer.

Reviewer #3

(Remarks to the Author)

REVIEWER COMMENTS

Reviewer #1 (Remarks to the Author):

In this paper, the authors use a multiplexed barcoding approach to assess the fitness of several metabolic salmonella mutants in various mouse models with altered microbiota. The authors find that some mutants have fitness defects across all mouse models, while others are only apparent in some mouse models. Overall this paper is a very deep analysis of carbohydrate utilization requirements for Salmonella and will be of interest to those who study Salmonella or carbohydrate utilization of other enteric pathogens. In my opinion no further experiments are required for publication, but there is an important experimental group missing (unperturbed C57BL/6J mice and some parts of the paper are difficult to read, so I have a few suggestions:

Author response: Thank you for this evaluation. Indeed, unperturbed C57BL/6J mice would be an important group for studying the metabolic requirements of *S. Typhimurium*. However, in our facility, unperturbed C57BL/6J mice exhibit such a high level of colonization resistance, that only a fraction of animals is showing significant *S. Typhimurium* loads on day 1, and an even smaller fraction retaining *Salmonella* in the faeces on the subsequent days post infection. The bottleneck in these unperturbed C57BL/6J mice is so strong, that we will get random loss of most/all strains of our library, without obtaining quantifiable fitness data. This is the main reason for excluding this model in our present study. We have elaborated further on this issue and describe how we have modified the manuscript to explain this technical problem to the readers in our response to Comment 2.

1.Experiments in this paper are well controlled and I do not believe any additional experiments are necessary for publication. One suggestion – since the authors already have access to this data – is to plot the evenness (of WT) as the X axis and mutant fitness on the Y axis. I am very curious if the fitness defects of individual mutants generally correlates with the severity of the infection bottleneck.

Author response: Previously, it was established that a Shannon Evenness Score (SES) of 0.9 can be considered a rigorous cutoff for concluding stable colonization for barcoded *S. Typhimurium* populations³. However, an SES below 0.9 does not necessarily indicate that mutant fitness cannot be assessed at all. As requested, we plotted data from the cecum content of C57BL/6J (OligoMM¹²) mice on day 4 post-infection for a selection of mutants (Fig. 1). In general, in samples featuring lower SES (<0.9, indicating a more severe bottleneck), we tend to observe a greater spread of the Competitive index data. In the case of the $\Delta nagB$ mutant, this spread can extend in both directions. For the $\Delta nanK$ mutant, there is no clear correlation between SES and mutant fitness. In conclusion, the selection process for the SES of the 7 wild types is a conservative measure to ensure the highest quality of fitness data. However, this does not necessarily mean that fitness data with an SES below 0.9 is generally uninterpretable.

Fig. 1: Data from the cecum content of C57BL/6J (OligoMM¹²) mice on day 4 post-infection is shown. The Shannon Evenness Score (SES) is plotted on the y-axis, while the corresponding competitive index for each respective mutant is plotted on the x-axis.

2. One caveat of this study is that unperturbed C57BL/6J mice are not included in competition experiments. I know that including streptomycin is helpful to improve colonization, but I believe that the “no step” model is used by others in the field. It would strengthen the paper (but is not necessary for publication) to test whether even a few mutants are defective for colonization in WT unperturbed C57BL/6J mice. In the absence of additional experiments, please directly reference this caveat.

[REDACTED]

Fig 2: *S. Typhimurium* colonization of unperturbed C57BL/6J (SPF) mice. Data from Nguyen et al., 2024.

Author response: Thank you for this comment. This is indeed a challenge we are facing. The biggest issue with unperturbed C57BL/6J SPF mice is that only a small fraction show sufficient *Salmonella* loads ($>10^6$ CFU per gram of feces) at day 1 post-infection, as previously shown in Nguyen et al., 2024. On day 1 post-infection, only 12 out of 30 mice, and on day 2, only 3 out of 30 mice, showed *Salmonella* loads sufficient for competitive index calculation via qPCR⁴ (Fig. 2). This is the main reason we typically don't include the unperturbed C57BL/6J mouse model in our analysis, as we cannot achieve a reproducible *S. Typhimurium* infection that is sufficient to perform these mutant pool analyses. To emphasize this downside, we included this caveat in the introduction when we discuss the different mouse models utilized in our study:

Unperturbed C57BL/6J (SPF) mice with an intact complex microbiome were not used due to the inability of *S. Typhimurium* to reproducibly colonize this particular model. (L74-76)

3. Please include a summary figure that provides the competitive indices for all mutants in this study. I appreciate how each figure tells a smaller story, but I find myself flipping back and forth quite a bit as I am reading this paper, and reading the raw count data in the supplementary tables is difficult

Author response: Thank you for the suggestion. We have included a heatmap showing the competitive index for each mouse model and the ATCC14028S competition in **Supplemental Fig. S2, S3, and S5.**

4. Line 18 - Including the WISH acronym as the 3rd sentence in the abstract without explaining what it is may be difficult to follow for readers who are unfamiliar with the technique

Author response: Thank you for this comment. We adapted the sentence accordingly:

Using **neutral genetic barcodes**, we tested 35 metabolic mutants across five different mouse models with varying microbiome complexities, allowing us to differentiate between context-dependent and context-independent nutrient sources. (L19-L21)

5. Line “five different mouse models” – a brief clarification on what these are could help with understanding (e.g., they could be different strains of mice, different routes of infection, etc.)

Author response: Thank you for this comment. We adapted the sentence accordingly:

Using neutral genetic barcodes, we tested 35 metabolic mutants across five different mouse models **with varying microbiome complexities**, allowing us to differentiate between context-dependent and context-independent nutrient sources. (L19-L21)

6. Line 57 – Lack of insertions within some genes is certainly an issue for some TnSeq libraries, but I do not think that target genes being “not effectively deactivated” is generally believed in the field. Consider omitting this point or providing a reference to show ineffective inactivation of random Tn insertions.

Author response: Thank you for this comment. We adapted the sentence accordingly and omitted this statement (L56-L58).

7. Line 60 – WISH may not be known to others in the field at this point in the manuscript. Consider elaborating

Author response: Thank you for the comment. We have included an additional sentence explaining the WISH acronym:

Recently, the wild-type isogenic standardized hybrid (WISH)-tag, a neutral genetic barcode for tracking strain abundance, was established for *S. Typhimurium*⁵. (L60-L62)

8. Line 60 – While I appreciate that rational design of mutants is a good strategy, it seems a bit odd to frame these experiments as overcoming limitations to TnSeq. TnSeq and barcoded mutant screens are not very comparable since one a screen (forward genetics) and the other is not (reverse genetics).

Author response: We excluded the following sentence: “These technical hurdles hinder systematic comparisons between different mouse models. (L56-L58)”, to present the comparison between TnSeq and our approach in a more neutral manner.

9. Line 80 – the text mentions qPCR and amplicon sequencing, but Figure 4B shows only amplicon sequencing. I’m a bit confused as to how qPCR can be used here. Would the authors perform 35 qPCR reactions per sample?

Author response: Thank you for this comment. We omitted the reference to qPCR to avoid any misunderstanding:

We investigated 35 metabolic mutants using the WISH-barcoding approach⁵, which allows amplicon sequencing for quantification of the strain abundance (**Fig. 1B, Supplementary Table S1**). (L84-L86)

10. Line 91 – is the *hyb* mutant a positive (fitness defect) or negative (no fitness defect) control

Author response: Thank you for this comment. We added the specification that *hyb* was identified as important (showing a fitness defect) for colonizing LCM mice⁶. However, the *hyb* mutant has not been characterized in the streptomycin-pretreated models or OligoMM¹²:

Additionally, a Δ *hyb* mutant, encoding a hydrogenase important for utilizing hydrogen as an electron donor **in LCM mice**⁶ and finally a SL1344 wild type was also included. (L97-L99)

11. Line 106 – I am surprised that germ-free mice have severe bottlenecks. Is there a reference for this? Ref 41 does not appear to have germ free data. It also seems like Figure S1 shows less severe bottlenecks in GF mice at d2 (s1f) than OligoMM12 mice at d2 (s1j)

Author response: Please excuse that our wording has been unclear. Thank you for noticing this. Population bottlenecks specifically occur in the streptomycin-pretreated C57BL/6J SPF model, starting on day 3 post-infection, as previously shown³. Germ-free mice were mistakenly mentioned in this context. We omitted the mention of germ-free mice from this sentence and added the specification in a separate sentence:

The same 2-day infection protocol was also used for germ-free mice. (L115-L116)

12. Figure S1 – please indicate what “cc” is in the figure itself

Author response: We added the meaning of "cc" (cecum contents) in the legend.

13. Line 138 – please include a figure, or at least a supplementary figure, that shows these results. It is difficult to check raw count tables

Author response: We included a heatmap for each mouse model in the Supplementary Figures, displaying the median fitness for each day post-infection (**Supplementary Fig. S2 and S3**).

14. Line 182 – provide figure for “did not alter SL1344 loads”, since Figure 2i appears to only show general *Salmonella* loads. Please also specify here whether these experiments were done at the same total *Salmonella* dose as others.

Author response: Thank you for this comment. Figure 2i shows only the SL1344 mutant pool loads, which is why we conclude that ATCC14028S "did not alter SL1344 loads." We adjusted the y-axis title to avoid this misunderstanding. We included the following sentence and changes to specify that this experiment used the same total *Salmonella* dose:

Mice were infected with the same total *Salmonella* dose (5×10^7 CFUs) as in previous experiments. (L205-L206)

15. I would like the authors to elaborate more on what the rationale for using ATCC14028S is in the paragraph starting at line 178. Since experiments thus far use a mixture of strains, it seems that we already know that these mutants are important for “intraspecies competition”, as all experiments so far have been competitive infections with Salmonella

Author response: Thank you for this comment. We added a sentence establishing that ATCC14028S is a well-established competitor for SL1344 in our lab:

To identify carbohydrates that are important for *S. Typhimurium* intraspecies competition, we analyzed the SL1344 mutant pool in the presence of **an additional** niche competitor, *S. Typhimurium* strain ATCC14028S. **SL1344 can coexist with ATCC14028S during colonization due to differences in metabolic capacity, particularly in D-galactitol utilization, as previously shown.⁷ For this reason, we aimed to investigate whether ATCC14028s alters the gut-luminal environment in a way that affects the fitness of specific carbohydrate mutants within our SL1344 mutant pool.** (L196-L204)

16. Line 197 – the authors say that the ptsG mutant did not have a defect in the LCM model and cite ref 52, but could they elaborate on how comparable these different experiments are? They may not be comparable if they were very different experimental designs. I cannot evaluate since ref 52 does not appear to be accessible online

Author response: Thank you for the comment. The referenced paper has since been published, and the citation has been updated accordingly. This paper employed TnSeq to study *S. Typhimurium* colonization in the C57BL/6J (LCM) mouse model. During the 4-day infection period *ptsG* showed no significant fitness defect. We included this information in this sentence:

However, this differed in the gnotobiotic LCM mouse model, where a *ptsG* mutant did not display a significant colonization defect **during a 4-day infection period in a TnSeq colonization experiment**⁴. (L224-L226)

We are confident that the TnSeq data and our presented data can be compared, as we observe similar mutant fitness for *frd* and *dcuABC*, which were originally identified in a TnSeq experiment.⁸

17. Line 211. I don’t follow this logic. The quadruple mutant has a strong colonization defect, so this fact alone would suggest it causes less inflammation. I don’t think that adding the T3SS mutations onto the quadruple mutant shows that the quadruple mutant does not trigger inflammation. It also seems like 3d does show that inflammation is occurring the quadruple mutant in the same model, although less than WT

Author response: Thank you for raising this concern. Indeed, with the avirulent mutant, we cannot conclusively determine whether the quadruple mutant is displaced due to its inability to trigger inflammation or its inability to utilize key carbohydrates. Since Reviewer 2 made a similar point in Comment 5, we included the following sentence to specify this caveat:

However, we cannot determine whether the avirulent quadruple mutant is displaced because of its failure to trigger inflammation or its inability to utilize key carbohydrate sources ($\Delta 4$ avirulent; Fig. 3b). (L240-L242)

18. Please add arrows in 3g

Author response: We included an arrow to indicate the edema.

19. Line 245 – is this true? Presumably each mutant is only 1/~50 in the whole pool, so isn't each mutant only about 2% of the population?

Author response: Our text, unfortunately, has been ambiguous. Thank you for noticing this. In our SL1344 mutant pool, each strain should have an abundance of 2% in the inoculum. The 0.1% in the infection mixture refers to the upcoming catch-up experiment, which is independent of our previous mutant pool experiment. In this experiment, the wild type is diluted to 0.1% of the infection mixture, while the carbohydrate mutant constitutes the remaining 99.9% of the inoculum. This setup allows us to assess whether the wild type can catch up to the same level as the carbohydrate mutant by utilizing the niche that becomes available due to the mutation in a specific carbohydrate utilization pathway. To avoid this misunderstanding we adapted the affected section in the manuscript:

In a different experimental setup, to determine whether a carbohydrate serves as an exploitable niche for *S. Typhimurium* wild type, we can supply the carbohydrate mutant in large excess compared to the wild type (with the wild type comprising 0.1% of the infection mixture). This allows us to see if the wild type can efficiently utilize the niche created by the mutant and reach similar levels within days, as previously shown for D-galactitol utilization.⁷ (L286-L291)

20. Line 273 – I feel like this is the most important sentence in the results, so I suggest that it is restated in the abstract and intro.

Author response: Thank you for this reassuring comment. We agree that this summarizes a core finding and have modified the Abstract as suggested:

Results showed that *S. Typhimurium* uses D-mannose, D-fructose, and likely D-glucose as context-independent carbohydrates across all five mouse models. (L21-L23)

21. Figure 5 – the authors should mention the caveat that the SPF mice were not treated with streptomycin, which makes these results not comparable to other experiments in the paper

Author response: Thank you for this comment. Nguyen et al., 2024 provides the comparison between streptomycin-pretreated and unperturbed C57BL/6J mice⁴. We also re-plotted the free monosaccharide measurements of cecum contents from streptomycin-pretreated C57BL/6J SPF mice in Fig. 5a of the main text and Supplementary Figure S7. In general, streptomycin treatment leads to a significant increase in L-arabinose and D-xylose concentrations, while the levels of other monosaccharides remain similar⁴. Furthermore, we updated the results section to include the new data.

22. If possible, please include at least one example of where the carbohydrate concentrations (as quantified in figure 5) correlates with the fitness of the mutant across those 3 mouse models

Author response: Thank you for these suggestions. In our opinion, it is not feasible to attribute individual monosaccharide concentrations to a specific fitness defect in a metabolic mutant that lacks one particular carbohydrate utilization pathway. The primary reason is the context-dependency of these observations. For example, L-arabinose is highly abundant in germ-free and

gnotobiotic C57BL/6J (OligoMM¹²) mice. However, the *araB* mutant, which is important for L-arabinose degradation, does not exhibit a severe fitness defect.

Table 1: Competitive index data of *S. Typhimurium* Δ *araB*.

	C57BL/6J (germ free)				
	d1	d2	cc		
Δ araB	0.958543	0.954421	1.024722		
	C57BL/6J (OligoMM12)				
	d1	d2	d3	d4	cc
Δ araB	0.88207	0.861883	0.861979	1.120294	1.002679

A second example is the *S. Typhimurium* Δ *galK* mutant, which is important for D-galactose degradation. D-galactose is similarly abundant in germ-free and OligoMM¹² C57BL/6J mice (Fig. 5B); however, the single mutant shows a fitness defect in germ-free mice but not in OligoMM¹² mice (Tab. 2; Fig. S2, Fig. S3). The *galK* mutation exhibits a fitness defect only when combined with a Δ *manA* Δ *fruK* background, as demonstrated by the triple Δ *manA* Δ *fruK* Δ *galK* mutant in C57BL/6J OligoMM¹² mice (Main text Fig. 3c).

Table 2: Competitive index data of *S. Typhimurium* Δ *galK*.

	C57BL/6J (germ free)				
	d1	d2	cc		
Δ galK	0.61805	0.469222	0.562591		
	C57BL/6J (OligoMM12)				
	d1	d2	d3	d4	cc
Δ galK	1.293026	1.846072	2.111085	2.685514	2.849297

In conclusion, *S. Typhimurium* appears to exhibit a mixotrophic lifestyle during colonization, facing a highly dynamic environment with constantly changing nutrient availability. For this reason, we believe that it is not possible to attribute a fitness defect in a mutant solely to the abundance (or concentration) of a specific monosaccharide.

23. Line 301 – while I definitely appreciate the phylogenetics, this section seems a little underbaked compared to the rest of the paper, especially for being at the end. For example, are these genes present in other organisms that do not live in the gut? Not a requirement to publish, but I would encourage the authors to consider changing its location in the manuscript, deepening the analysis, keeping it only as discussion, or omitting it entirely.

Author response: Thank you for mentioning this concern. *Salmonella*, *Shigella*, *Citrobacter*, and *Escherichia* are commonly used genera in *in vivo* microbiology with important growth phases in the animal gut lumen. Our intention was to examine how widely these carbohydrate degradation systems are distributed among these closely related genera. Indeed, looking at D-glucose, D-mannose, D-fructose, etc., and observing that these metabolic systems are present in almost 100% of the analyzed genomes may not seem surprising, as D-glucose degradation, in particular, is even found in highly reduced symbiont genomes⁹. However, in our opinion, this is an important point currently missing in the field: that these monosaccharides form the basis of colonization. Considering recent publications that underscore the importance of mixed acid fermentation for *Salmonella*^{4, 10}, these monosaccharides are the missing piece that drives the initial growth of

Enterobacteriaceae, through monosaccharide-fueled mixed acid fermentation. Fig. 5b allows us to illustrate this point. However, to test the hypothesis that context-dependent carbohydrate utilization systems may tend to belong to the accessory genome, we extended this analysis to include D-xylose (*xyl* operon), hydrogen (*hyb* operon), L-idonate (*idn* operon), and D-galactonate (*dgo* operon). Hexuronate and hydrogen utilization are associated with the accessory genome in non-typhoidal *Salmonella*, *Escherichia*, *Shigella*, and *Citrobacter*, whereas D-xylose and hydrogen utilization is associated with the core genome (**Supplementary Fig. S8, Table S8**). In particular, for D-xylose, this highlights an important observation: D-xylose is relatively abundant in the gut (**Fig. 5a**); however, *xylB* shows no fitness loss (**Supplementary Fig. S2, S3**), and D-xylose does not appear to be an exploitable niche (**Fig. 4b, see main text L356-L366**). These kinds of observations can provide hints that *S. Typhimurium* also has an intricate carbon utilization hierarchy in the gut, similar to what has been studied in great detail for *E. coli* in vitro ¹¹. In conclusion, we would like to keep this section and hope that this extended analysis suffices.

24. Figure legends – please include how many replicates are used whenever boxplots are displayed (e.g. figure 2a)

Author response: Thank you for mentioning this. We included the sample size in either the respective subpanel or figure legends, and in most figures, we also included the individual data points.

Reviewer #2 (Remarks to the Author):

This study examines the carbohydrate sources required for *Salmonella* to colonize animals. It uses several murine models, including those with complex but disrupted intestinal microbiota, and those with defined microbiota, as well as germfree mice. Using uniquely tagged strains in competition, the authors assess survival by competition assays to identify the important pathways for carbohydrate utilization. They present evidence that D-glucose, D-mannose, D-fructose, and D-galactose are used in all models tested, with other carbohydrates being important only under specific conditions, suggesting differences in carbohydrate utilization and productions among them. The work described was well conceived and carefully performed to reduce bias. It has resulted in new and interesting information. Some of the conclusions drawn, however, are not well explained or supported. Specific comments:

Author response: Thank you for this overall positive assessment and for taking the time to provide valuable comments.

1. Line 82: The dilutions made using the wild-type strain are not well described.

Author response: Thank you for phrasing this concern. We slightly adjusted the corresponding section:

The inoculum comprised four groups: control mutants with known colonization defects, seven wild-type strains to assess the evenness of distribution, a wild-type dilution series **to establish the window of measurement**, and 35 metabolic mutants deficient in carbohydrate-utilizing enzymes (**Fig. 1c**). (L86-L89)

Furthermore, we revised the concluding section of the corresponding paragraph:

A wild-type dilution series established the measurement window for each mouse and time point, **typically displaying a difference of 10^{-3}** , ensuring accurate quantification **within this limit of detection.** (L101-L104)

We have a section on the dilution standards in the Materials and Methods, providing technical details and referencing a plotted example from streptomycin-pretreated C57BL/6J mice. In addition, we also included a plotted one example of the gnotobiotic C57BL/6J (OligoMM¹²) mouse model:

In the streptomycin-pretreated C57BL/6J and the gnotobiotic C57BL/6J (OligoMM¹²) mouse model, wild-type dilutions at 10^{-3} were consistently detectable (Supplementary Fig. S8b and S8c). This trend was consistent across all tested models. Based on the minimum number of WISH counts (≥ 10) and the dilution standard, the competitive index of mutants that were below the limit of detection were conservatively set to a competitive index of 10^{-3} . (L570-L574)

2. Line 150: The *galK* mutant was found to be deficient in only germfree and streptomycin-treated mice. How can this be explained? One hypothesis would be that some specific constituents of the microbiota produce galactose, but this does not appear to be true (Fig. 5a). What model can be proposed to explain this finding?

Author response: Thank you for this comment. Indeed, the *galK* mutant showed context-dependent importance in the germ-free and streptomycin-pretreated mouse models (main text Fig. 2a-e). However, when combined with *manA* and *fruK* mutations, a fitness defect was observed in the OligoMM¹² model (main text Fig. 3c). D-galactose concentrations are higher in germ-free and streptomycin-pretreated animals (main text Fig. 5a)⁴. D-galactose utilization is subject to catabolite repression¹². One possible explanation could be that catabolite repression and the availability of monosaccharides place galactose lower in the hierarchy, making it less important in these environments when D-fructose, D-mannose, or D-glucose are available.

S. Typhimurium appears to have a mixotrophic lifestyle in the gut, encountering a varying nutritional environment at different time points. Since multiple carbohydrates are used at once, the importance of D-galactose may change in a highly context-dependent manner. Also, catabolite repression is known to enable *E. coli* to selectively and sequentially utilize carbohydrates, based on key regulatory systems¹¹. This is, of course, highly speculative, as there is no literature that provides insight into the importance of catabolite repression and the sequential use of carbon sources in an in vivo setting. Since this matter is currently too speculative, we feel that it would be pre-mature to engage in this discussion in the current paper.

3. Line 159: The *nagB* mutant has a different, but also inconsistent phenotype: deficient in germfree and low-complexity microbiome mice, but not in those treated with streptomycin. Explanations for this?

Author response: Thank you for this comment. Interestingly, N-acetylglucosamine and D-glucosamine concentrations are relatively low in LCM mice⁴, OligoMM¹² mice and streptomycin pretreated models, compared to germ-free mice (Fig. 5a). However, the *nagB* mutant still shows the most severe fitness defects not only in germ-free, but also in LCM mice. Similar to your second comment, we can only speculate on the underlying factors that contribute to this observation. However, we would argue, similar to the *galK* phenotype, that context-dependent effects related to diverse changes in carbohydrate concentrations or catabolite repression and the sequential use of carbon sources will most likely play a role.

4. Line 167: The phenotype of the *kdgK* mutant appears to be similar to that of the *galK* mutant. In this case, “post-antibiotic stress” is cited as a possible cause. In germfree mice, however, no antibiotic has been used. How would this apply to germfree animals?

Author response: Thank you for this comment. At this stage, we really don't know. All four sugar acids show a trend towards higher concentrations in germ-free and streptomycin pretreated SPF mice, compared to LCM mice⁴, OligoMM¹² mice or unperturbed C57BL/6 (SPF) mice⁴ (Fig. 5a and data in reference⁴). While we do not have an explanation for this, earlier work has proposed that an inflammatory response might lead to sugar acid formation in the gut of streptomycin pretreated mice¹³. In this publication, they also observed sugar acids in germ-free mice but could not provide an explanation for this. Further work will be required to assess why germ-free mice tend to show elevated sugar acid concentrations, even without antibiotic treatment. To highlight this knowledge gap for readers, we have modified the text.

Previous studies suggested that post-antibiotic stress creates a hexuronate-rich niche for *S. Typhimurium* colonization by oxidizing hexoses such as D-glucose and D-galactose¹³, indicating a general mechanism that likely also oxidizes D-fructose and D-mannose. Future research should investigate the mechanisms that lead to an increase in hexuronates in the gut lumen of germ-free mice. Regardless, the $\Delta kdgK$ phenotypes are also consistent for $\Delta uxaC$ (D-galacturonate) mutants, which show consistently lower competitive indices in streptomycin-pretreated models than in gnotobiotic models (Supplementary Fig. S4a). (L182-L189)

5. Line 209: An argument is made that the *manA*, *fruK*, *galK*, *ptsG* mutant is incapable of triggering inflammation because its colonization is indistinguishable from that of an isogenic mutant with disruptions of the type III secretions systems required to produce intestinal inflammation (Fig. 3B). It seems equally plausible that this quadruple mutant can induce inflammation but fails to survive because of its inability to utilize important carbohydrates. Inflammation could have been directly tested, using the methods of Fig. 3e-g, but the two experiments used different mouse models, and the avirulent mutant was not tested for its inflammatory effects.

Author response: Thank you for this comment and observation. Indeed, we cannot definitively determine from this experiment whether the avirulent quadruple mutant is displaced due to its failure to trigger inflammation or its inability to utilize key carbohydrate sources. For this reason, we included this limitation in a subsequent sentence:

However, we cannot determine whether the avirulent quadruple mutant is displaced because of its failure to trigger inflammation or its inability to utilize key carbohydrate sources ($\Delta 4$ avirulent; Fig. 3b). (L240-L242)

We specifically tested the avirulent strain in our streptomycin-pretreated C57BL/6J mice because we knew from a previous study that avirulent *S. Typhimurium* can be significantly displaced by the regrowing microbiota (Fig. 3).

[REDACTED]

Figure legend taken from Stecher et al, 2007²

However, this is not the case for avirulent *S. Typhimurium* in the OligoMM¹² model, where both avirulent and wild-type *S. Typhimurium* are capable of stably colonizing.

[REDACTED]

Figure 4. *S. Typhimurium* wild type (closed circles) and avirulent (open circles) infection of OMM12 mice over the course of 4 days post infection. Figure taken from Beutler et al 2024¹

For these reasons we only used the avirulent $\Delta manA \Delta fruK \Delta galK \Delta ptsG$ quadruple mutant in the streptomycin pretreated C57BL/6J model.

6. Line 237: The inflammatory markers (Fig. 3e-f) did not differ between the wild type and the fruK, galK,ptsG mutant. Only the additional loss of manA produced a difference. Does this implicate D-mannose specifically as the important carbohydrate? The manA mutant alone is deficient in both colonization and competitive index (Fig. 3b-c).

Author response: Thank you for raising this interesting point. Based on these observations, one could indeed conclude that D-mannose utilization may play a critical role in this particular mouse model. However, our aim was to avoid establishing a clear hierarchy. This would be better compatible with carbohydrate co-use. Instead, we focused on clustering carbohydrate sources into context-dependent and context-independent groups, considering factors such as monosaccharide abundance, mutant fitness, and niche exploitability. Furthermore, *manA* is an important isomerase that reversibly converts fructose 6-phosphate to mannose 6-phosphate. Mannose 6-phosphate is particularly used in the biosynthesis of GDP-mannose and GDP-fucose, which are potential components of the O-antigen. Since the D-mannose transporter is intact and there is an abundance of D-mannose in the environment, the supply of mannose 6-phosphate for biosynthesis should not be an issue. In conclusion, we chose to interpret the fitness data conservatively, limiting our statements primarily to whether a metabolic gene is important or not important.

7. Line 301: The genes shown in this study to be important for Salmonella colonization are demonstrated also to be widely present in members of the Enterobacteriaceae. This evidence is used to argue for their broader importance for survival in animals. If correct, one might also expect the converse to be true: pathways not important for colonization would be less widely shared. Is this the case? Several genes used in this study could be tested, including *kdgK*, *xylB*, *hyb*, *idnD*, and possibly *uxaC*.

Author response: Thank you for this comment. To address this important question, we extended our analysis to determine if the aforementioned genes are associated with the core genomes of *Salmonella*, *Escherichia*, *Citrobacter*, and *Shigella*. We also included the requested genes in our analysis and added a corresponding section in the results:

To test the hypothesis that context-dependent carbohydrate utilization systems may tend to belong to the accessory genome, we extended this analysis to include D-xylose (*xyl* operon), hydrogen (*hyb* operon), L-idonate (*idn* operon), and D-galactonate (*dgo* operon). Hexuronate utilization are associated with the accessory genome in non-typhoidal *Salmonella*, *Escherichia*, *Shigella*, and *Citrobacter*, whereas D-xylose and hydrogen utilization is associated with the core genome (Supplementary Fig. S8, Table S8). Interestingly, D-xylose is relatively abundant in the cecum contents of mice (Fig. 5a); however, it does not appear to serve as a nutrient source for *S. Typhimurium* during colonization (Fig. 4b and Supplementary Fig. S2, S3). (L356-L366)

Indeed, one might assume that metabolic genes are lost when the corresponding carbohydrate is not abundant in the bacterium's environment. However, gene loss represents the extreme case of genetic adaptation to a specific environment. Recent publications have highlighted that even a single nucleotide mutation¹⁴ or point mutation in the primary sequence¹⁵ is sufficient to alter the expression of a metabolic system or silence it entirely. This provides the bacterium with enough flexibility to adapt to different environments without the need to lose the gene. In conclusion, we agree that there is likely a correlation between a gene being less widely shared and the corresponding monosaccharide being less abundant.

8. Comparing the results of Fig. 2e and Fig. 3c, the *galK* mutation produces no effect singly but demonstrates one only when combined with *manA*, *fruK*, and *ptsG* mutations. Does this indicate that galactose is not a preferred energy source for Salmonella in the intestine and is used only when other more favorable sources are not available?

Author response: Thank you for this comment. As mentioned earlier in the response to comment 2, this would also be our interpretation of the data. A possible explanation, as previously mentioned, could involve catabolite repression, which influences the sugar utilization hierarchy. Since *galK* is not part of the phosphotransferase system, it is likely given lower priority than D-mannose and D-fructose. However, we chose not to delve into this level of detail because, at the current state of knowledge, we simply don't know how significant catabolite repression and the concept of sequential carbon utilization are for *Enterobacteriaceae* in colonizing the mammalian gut.

Reviewer #3 (Remarks to the Author):

Author response: Thank you for your time and effort in providing comments.

1. Beutler, M. *et al.* Contribution of bacterial and host factors to pathogen "bloating" in a gnotobiotic mouse model for *Salmonella enterica* serovar Typhimurium-induced enterocolitis. *Infect Immun* **92**, e0031823 (2024).
2. Stecher, B. *et al.* *Salmonella enterica* serovar typhimurium exploits inflammation to compete with the intestinal microbiota. *PLoS Biol* **5**, 2177-2189 (2007).
3. Maier, L. *et al.* Granulocytes impose a tight bottleneck upon the gut luminal pathogen population during *Salmonella typhimurium* colitis. *PLoS Pathog* **10**, e1004557 (2014).
4. Nguyen, B.D. *et al.* *Salmonella Typhimurium* screen identifies shifts in mixed-acid fermentation during gut colonization. *Cell Host Microbe* (2024).
5. Daniel, B.B.J. *et al.* Assessing microbiome population dynamics using wild-type isogenic standardized hybrid (WISH)-tags. *Nat Microbiol* **9**, 1103-1116 (2024).
6. Maier, L. *et al.* Microbiota-derived hydrogen fuels *Salmonella typhimurium* invasion of the gut ecosystem. *Cell Host Microbe* **14**, 641-651 (2013).
7. Gül, E. *et al.* Differences in carbon metabolic capacity fuel co-existence and plasmid transfer between *Salmonella* strains in the mouse gut. *Cell Host & Microbe* (2023).
8. Nguyen, B.D. *et al.* Import of Aspartate and Malate by DcuABC Drives H(2)/Fumarate Respiration to Promote Initial *Salmonella* Gut-Lumen Colonization in Mice. *Cell Host Microbe* **27**, 922-936 e926 (2020).
9. Hutchison III, C.A. *et al.* Design and synthesis of a minimal bacterial genome. *Science* **351**, aad6253 (2016).
10. Rogers, A.W.L. *et al.* *Salmonella* re-engineers the intestinal environment to break colonization resistance in the presence of a compositionally intact microbiota. *Cell Host Microbe* (2024).
11. Okano, H., Hermsen, R., Kochanowski, K. & Hwa, T. Regulation underlying hierarchical and simultaneous utilization of carbon substrates by flux sensors in *Escherichia coli*. *Nat Microbiol* **5**, 206-215 (2020).
12. Semsey, S., Krishna, S., Sneppen, K. & Adhya, S. Signal integration in the galactose network of *Escherichia coli*. *Molecular microbiology* **65**, 465-476 (2007).
13. Faber, F. *et al.* Host-mediated sugar oxidation promotes post-antibiotic pathogen expansion. *Nature* **534**, 697-699 (2016).
14. Cherrak, Y. *et al.* Non-canonical start codons confer competitive advantage in carbohydrate utilization for commensal *E. coli* in the murine gut. *Nature microbiology*, accepted (2024).

15. Machado, L.F.M. & Galán, J.E. Loss of function of metabolic traits in typhoidal without apparent genome degradation. *Mbio* **15**, e00607-00624 (2024).